# HIV-Tocky system to visualize proviral expression dynamics
Omnia Reda [1,2], Kazuaki Monde [3], Kenji Sugata[1], Akhinur Rahman[1], Wajihah Sakhor[1], Samiul Alam Rajib [1], Sharmin Nahar Sithi [1], Benjy Jek Yang Tan [1], Koki Niimura[4], Chihiro Motozono [5], Kenji Maeda[6], Masahiro Ono [7], Hiroaki Takeuchi [8] & Yorifumi Satou [1] ✉

Determinants of HIV-1 latency establishment are yet to be elucidated. HIV reservoir comprises a rare fraction of infected cells that can survive host and virus-mediated killing. In vitro reporter models so far offered a feasible means to inspect this population, but with limited capabilities to dissect provirus silencing dynamics. Here, we describe a new HIV reporter model, HIV-Timer of cell kinetics and activity (HIV-Tocky) with dual fluorescence spontaneous shifting to reveal provirus silencing and reactivation dynamics. This unique feature allows, for the first time, identifying two latent populations: a directly latent, and a recently silenced subset, with the latter having integration features suggestive of stable latency. Our proposed model can help address the heterogeneous nature of HIV reservoirs and offers new possibilities for evaluating eradication strategies.

HIV continues to be an unresolved global public health concern[1] and is one of the three world's deadliest infections[2]. Combined anti-retroviral therapy (cART) showed an incredible effect in turning HIV-1 infection from a life-threatening infectious disease to a chronic viral infection. However, cART is unable to fully eradicate the virus from infected individuals[3] because it is not cytotoxic on virally infected cells and does not block clonal expansion or provirus expression[4].

During the acute phase of HIV-1 infection and prior to cART initiation, most infected cells expressing viral antigens will be eliminated either by host immune clearance or by the viral cytopathic effect (CPE)[5]. Simultaneously, a reservoir is established through a small pool of stably integrated proviruses in the host genome of infected cells that can escape CPE and immune clearance with the eventual support of host cell survival and clonal expansion[6–8]. This reservoir contracts only mildly even after years of anti-retroviral therapy (ART) and holds a threatening reactivation capacity once ART is interrupted[3,9].

HIV-1 reservoir is heterogeneous in terms of reservoir site, provirus intactness, and replication fitness. The reservoir is mainly composed of long-lived resting memory CD4 + T cells, while macrophages, microglia, and dendritic cells can also contribute to specific anatomical sanctuaries[10,11]. Defective proviruses are predominant in vivo[12], however, not all intact integrated proviruses are inducible[13]. Although this may be partially

explained by their stochastic inducibility[12–14], their reactivation dynamics are unclear. Furthermore, integrated proviruses from ART-treated individuals are not uniformly transcriptionally silent, and their individual transcriptional status is governed by interlacing transcriptional, post-transcriptional, and epigenetic mechanisms operating at provirus integration sites[15–18]. This convoluted ability of the HIV-1 provirus to hide in latent reservoirs represents the principal obstacle to an HIV-1 cure[3,19–21].

Interrogation of this integrated reactivatable pool of proviruses has relied on in vitro assays for long owing to its rarity. Recent advances in multi-omics analysis have revolutionized the ability to examine this pool in people living with HIV (PLWH) ex vivo through which crucial information on the determinants of its maintenance has been obtained. The outstanding key question is how exactly the small pool of reservoir cells is initially selected to survive among these extremely heterogeneous infected cells.

Meanwhile, utilizing HIV-1 in vitro reporter models has widely aided the advents in eradication strategies. We previously established the widely distributed provirus elimination assay (WIPE assay) as an in vitro HIV infection model with hundreds of infected clones that can be maintained for several months[22]. This model recapitulated the complexity of infected cells reported for in vivo HIV-1 infection, as well as the persistence of defective HIV-1 proviruses. However, the evaluation of provirus expression using the

[1]Division of Genomics and Transcriptomics, Joint Research Center for Human Retrovirus Infection, Kumamoto University, Kumamoto, Japan. [2]Microbiology Department, High Institute of Public Health, Alexandria University, Alexandria, Egypt. [3]Department of Microbiology, Faculty of Life Sciences, Kumamoto University, Kumamoto, Japan. [4]School of Medicine, Kumamoto University, Kumamoto, Japan. [5]Division of Infection and Immunology, Joint Research Center for Human Retrovirus Infection, Kumamoto University, Kumamoto, Japan. [6]Division of Antiviral Therapy, Joint Research Center for Human Retrovirus Infection, Kagoshima University, Kagoshima, Japan. [7]Department of Life Sciences, Imperial College London, London, UK. [8]Department of High-risk Infectious Disease Control, Tokyo Medical and Dental University (TMDU), Tokyo, Japan. ✉e-mail: y-satou@kumamoto-u.ac.jp

WIPE assay model requires cell lysis or permeabilization to quantify the viral RNA or protein.

Several generations of dual-fluorescent viruses utilizing green fluorescent protein (GFP) as a reporter for productive infection have been developed. They offered useful information on the early silencing of the HIV-1 provirus shortly after infection[6,23], the in vitro provirus inducibility in response to various latency-reversing agents (LRAs)[24], and addressed the convenience of the shock and kill strategy[25–28]. Despite that, GFP is a stable fluorescent protein with a long half-life[29], which results in a gap between the timing of HIV-1 expression and long-term GFP positivity, which in turn limits the temporal resolution of GFP in capturing provirus dynamics.

To overcome difficulties in analyzing temporal dynamics of transcription in vivo, recently a breakthrough was made in immunology by establishing a Timer of cell kinetics and activity (Tocky). This Tocky model uses a fluorescent Timer protein that spontaneously changes its emission spectrum from blue to red upon maturation of blue chromophore by oxidation[30,31]. In this study, aiming at the simultaneous visualization of provirus expression dynamics, provirus intactness, and integration landscape, we propose a recombinant HIV construct equipped with this fluorescent Timer protein (HIV-Tocky). We showed that by synchronizing its emission spectrum spontaneous change to provirus expression, Timer fluorescence can punctually trace provirus temporal dynamics from expression to silence. We investigated the integration characteristics of different provirus expression statuses. Moreover, we traced infected cells for clone formation and probed their response to potential latency-promoting agents (LPAs). This in vitro model for tracing real-time provirus kinetics provides a substantial change in the way latent reservoir composition is described and further assessed for eradication strategies.

## Results

### Establishment of HIV-Tocky system equipped with Fluorescent Timer reporter to capture provirus transcriptional dynamics

Previously, the Ono group developed the Tocky model as a new tool for analyzing transcriptional dynamics using a Fluorescent Timer protein[31], whose emission spectrum spontaneously changes from blue to red fluorescence following chromophore maturation[30]. As the maturation half-life of the Timer-blue chromophore is 4 h and the mature Timer-red protein is stable with a half-life is 122 h[32], the Tocky system allows the analysis of the dynamics of gene expression and reactivation. Thus, to analyze HIV-1 proviral expression dynamics, we designed a recombinant HIV-1 NL4-3-based molecular clone[33] harboring the coding sequence of Timer protein in the *nef* region, where Timer expression is under the control of the HIV-1 promoter in the 5' long terminal repeat (LTR) ; (HIV$_{Timer}$) (Fig. 1a; upper panel). The concept of utilizing the temporal shifting feature of Timer fluorescence to visualize HIV-1 proviral expression is best understood by describing the possible fates of the host cell after infection. Infected cells can have one of the following two immediate fates: firstly, an infected cell will actively transcribe mRNAs from the integrated provirus harboring the Fluorescent Timer protein (Timer-FP), rendering it positive for blue fluorescence but still negative for red fluorescence (B + , early). Secondly, infected cells will directly become latent, with no or sub-threshold provirus expression and without any detectable Timer expression (TN). Proviruses within the B+ population can either continue to be transcribed or stop expressing. This creates a mixed population with double positivity for blue and red fluorescence (B + R + , persistent). Here it is considered that most Timer-positive cells will be eliminated by virus-induced CPE and only a small fraction of the B+ or B + R + cells that have escaped CPE and stopped transcription will eventually lose Timer-Blue fluorescence and acquire the red fluorescence forming the (R + ) population. Figure 1b shows a conceptual scheme for the proviral expression dynamics in a single-round infection HIV-Tocky model. To establish a proof-of-concept for the HIV-Tocky system, we infected Jurkat T cells with HIV$_{Timer}$ and analyzed Timer expression dynamics by flow cytometry (Fig. 1c and Supplementary Fig. 1a). Timer fluorescence was first detectable at 24 h post-infection and evolved in a fan-like movement from B+ to R + , as previously demonstrated by the

Nr4a3-Tocky system, at which Timer protein is inducible upon T cell stimulation[30]. The R+ population, representing proviruses with recently arrested transcription, started to be evident from 3 days post-infection (dpi) and accumulated gradually over time. Timer-FP positivity peaked at 3 dpi and decreased over time, presumably due to CPE and/or cell cycle arrest induced by viral protein expression in B+ and B + R+ cells (Fig. 1e; left panel). Next, to identify infected cells in an LTR-independent manner, we modified HIV$_{Timer}$ by inserting the expression cassette of nerve growth factor receptor lacking the intracellular domain (ΔNGFR) in the *nef* region (HIV$_{TNGFR}$, Fig. 1a; lower panel). NGFR expression in HIV$_{TNGFR}$ is driven by the constitutively active promoter of *Eukaryotic Translation Elongation Factor 1 Alpha 1(EEF1A1)*[34], and upon provirus integration, NGFR will be expressed on the surface of infected cells enabling the identification of infected cells by flow cytometry irrespective of provirus expression. Infected cells with no proviral expression will be positive only for NGFR but negative for Timer-FP. We infected Jurkat T cells with the pseudotyped single-round HIV$_{TNGFR}$ virus. Total cells were first gated for NGFR expression as a marker of infection, and then Timer-FP positivity among infected cells (NGFR + ) was analyzed by flow cytometry over time (Fig. 1d and Supplementary Fig. 1b). We found that the pattern of Timer-FP expression was similar to that of HIV$_{Timer}$ infection and that only a small percentage of infected cells expressed the provirus at all time points, even in the highly proliferating Jurkat T cells (Fig. 1e, right panel, and Fig. 1f), which is consistent with a previous report on primary CD4 + T-cells' infection[25]. We next isolated primary CD4 + T cells from healthy donors' PBMCs and infected them with HIV$_{Timer}$. The fan-like movement of Timer fluorescence from B+ to R+ that has been demonstrated in Jurakt T cell infection was similarly observed in primary CD4 + T cell infection (Fig. 1g). To evaluate the ability of the HIV-Tocky system to capture provirus dynamics in different cell lineages, we infected THP-1 monocytic cells with the same recombinant viruses. Similar to Jurkat T cells, the Timer-Blue fluorescence appeared 24 h after infection, and R+ cells peaked at 4 dpi (Supplementary Fig. 2a). Timer-FP-expressing cells constituted a small proportion of all infected cells and gradually decreased over time (Supplementary Fig. 2b). To confirm whether Timer expression is correlated with viral gene expression, we sorted each Timer fraction after infecting Jurkat T cells with HIV$_{TNGFR}$ (Supplementary Fig. 3a) and quantified viral unspliced and multiply spliced mRNA using reverse transcription polymerase chain reaction (qRT-PCR). To obtain enough cells for downstream analysis, TN, B + , B + R + , and R + cells were sorted from 1, 2, 3, and 4 dpi samples, respectively. The level of early produced and multiply spliced Tat/Rev transcripts was the highest in B + cells and the lowest in R+ cells. The level of late unspliced Gag transcripts was similar between the B+ and B + R+ populations but low in the R+ population (Fig. 1h). To rule out the possibility that the provirus silencing monitored by Timer expression was attributed to general transcriptional shutdown rather than specific HIV promotor silencing, we quantified NGFR transcript levels across different Timer populations. NGFR mRNA showed constant expression levels regardless of the provirus expression status (Supplementary Fig. 3b). These results shows that Timer expression successfully captures viral gene expression dynamics. Furthermore, we analyzed the copy number of cell-associated viral DNA by droplet digital PCR (ddPCR). While there were approximately 1.6 copies of *Gag* DNA observed per infected cell (Supplementary Fig. 4a, b, c), 2 circular LTR (2-LTRc) DNA, a form of unintegrated viral DNA peaked at 1 dpi with 21 copies/ul. We calculated a percentage of 2-LTRc contribution to all *Gag* copies detected over the first 3 days of infection. 2-LTRc constituted less than 1% of all *Gag*-detected copies (Supplementary Fig. 4d, e, f). This result suggests that most of the cell-associated viral DNA consisted of integrated proviruses. In addition to that, we characterized the structure of the HIV-1 proviral genome by nearly full-length PCR, using a single copy of the HIV-1 genome as a template[35]. Of the analyzed 27 proviruses from TN, B + , and B + R+ Timer fractions, more than 90% were near full-length copies (Supplementary Fig. 4g). Collectively, these data indicate the successful establishment of the HIV-Tocky model that can capture the HIV-1 provirus dynamics of reactivation and silencing.

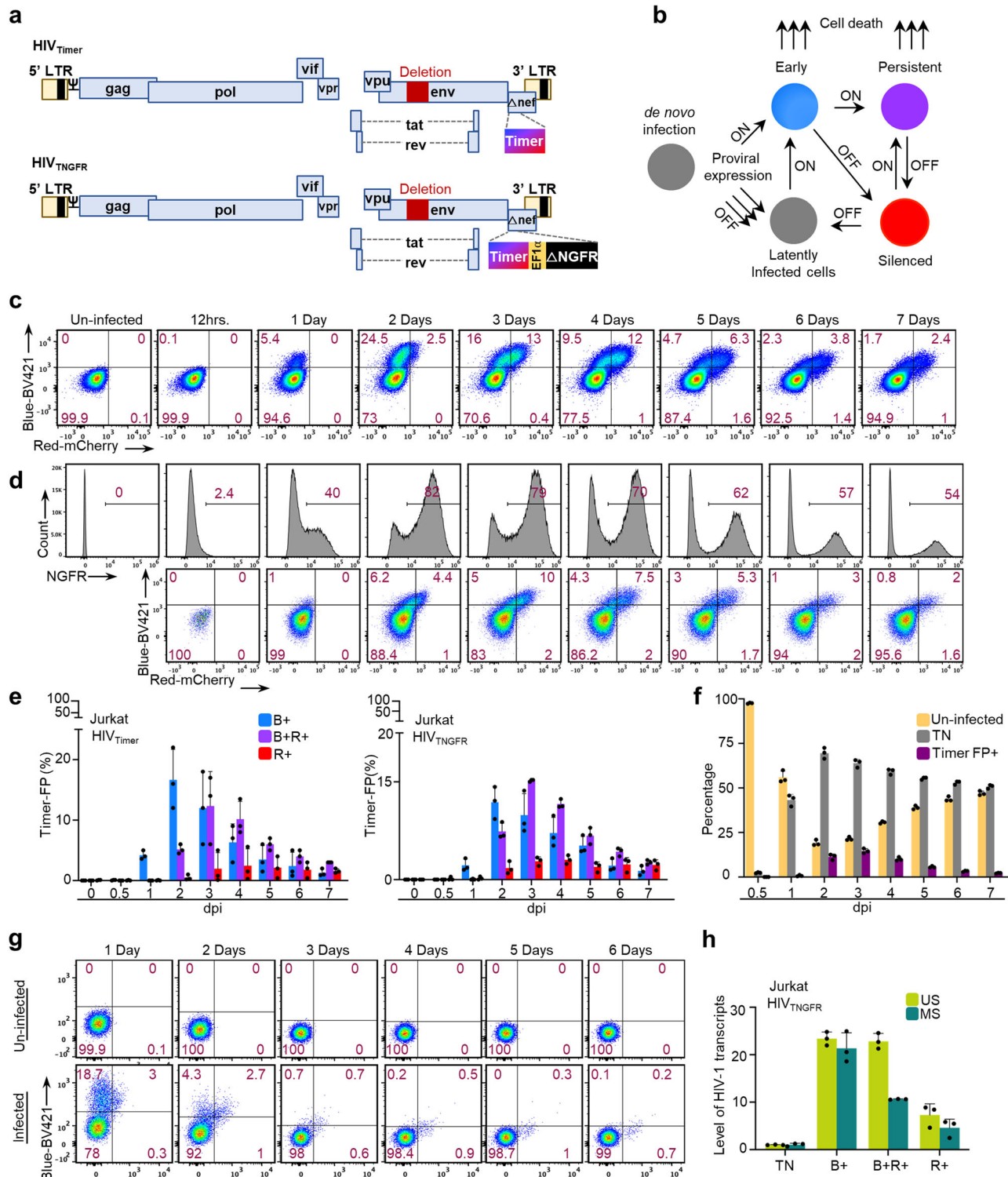

## Provirus integration site analysis of Jurkat T cells infected with HIV-Timer virus

The interplay between genetic and epigenetic circumstances at the HIV-1 provirus integration site impacts its fate of either expression or silencing[26,36–43]. Using the HIV-Tocky system, we aimed to analyze the relationship between HIV integration sites (ISs) and the categorized provirus transcriptional status by Timer-FP. As described above, the B+ cell population is composed of infected cells with heterogeneous fates, including continuous expression, apoptosis, or "to-be-latent", whereas the R+

population is homogenously enriched with infected cells harboring the proviruses that were once expressing and later their expression was terminated. Thus, to investigate latency-initiating circumstances, we compared the integration environment between R+ and other Timer populations by sorting TN, B+, B+R+, and R+ populations (Supplementary Fig. 5) and performing IS analysis. HIV ISs were determined by amplifying the junction between the HIV-1 LTR and the flanking host genome using ligation-mediated PCR as previously described[44]. We obtained 12,708 unique integration sites, including 4750, 3548, 3919, and 491 ISs in the TN, B+,

**Fig. 1 | Establishment of in vitro HIV-Tocky reporter system to monitor provirus expression dynamics. a** Schematic representation of HIV-1-Timer constructs (top: HIV$_{Timer}$) and (bottom: HIV$_{TNGFR}$). **b** Schematic for Timer protein expression during different phases of provirus expression or silencing. **c** Representative flow plots from time course infection of HIV$_{Timer}$ in Jurkat T cells. Jurkat cells were infected by the HIV$_{Timer}$ virus adding and followed up for Timer expression until 7 dpi. The plot is representative of three independent experiments. **d** Jurkat T cells infection by HIV$_{TNGFR}$. Histogram plots representing ΔNGFR percentages until 7 dpi (upper panel). Representative Timer-axis shift by flow plots from time course infection (lower panel). The plots are representative of three independent experiments. **e** Representative bar graphs denoting the percentage of each Timer population from infection experiments in (**c**) (left panel) and in (**d**) (right panel). For each graph (n = 3 biologically independent experiments, mean ± SD). **f** Bar graph

representing Timer FP+ fraction (in magenta) in comparison to TN (gray) or uninfected cells (yellow) across infection time points from (**d**). **g** Representative flow plots from time course infection of Primary CD4 + T cells by HIV$_{Timer}$. Primary CD4 + T-cells were isolated from PBMCs by negative selection and then activated with Dynabeads Human T-activator CD3/CD28 for 24 h. The Retronection-bound virus method was used for infection. Infected cells were followed up for Timer expression until 6 dpi. Gating for Timer quadrants in infected cells (lower panel) was performed according to un-infected controls (upper panel). **h** Total RNA isolated from each Timer population of Jurkat-infected cells with HIV$_{TNGFR}$ was subjected to SYBR green RT-qPCR analysis. Unspliced *Gag* (US) in teal and multiply-spliced *tat/Rev* (MS) in lime green. HIV-1 mRNAs were quantified relative to cellular 18s rRNA. (n = 3 biologically independent experiments, mean ± SD).

B + R + , and R+ populations, respectively. Initially, we investigated the distribution of HIV ISs across chromosomes and compared the data from this study with that from previously published reports, including in vitro infection (65,924 ISs) and peripheral blood mononuclear cells (PBMCs) isolated from HIV-infected individuals prior to ART initiation (13,142 ISs)[15]. First, we observed a similar enrichment of integration in the chromatin of gene-dense chromosomes 17 and 19 in all three datasets (Fig. 2a; upper panel). Second, HIV ISs were similarly enriched in the genic region of the host genome in all three datasets (Fig. 2b; upper panel). These data indicate that the insertion of the Timer-FP coding sequence in the HIV genome seems to have little impact on the distribution of HIV-ISs that were comparable to previous reports, both in vitro and in vivo.

Next, we analyzed the distribution of ISs across Timer-FP populations. Timer-FP ISs were widely distributed across all chromosomes (Fig. 2a, lower panel), with no distinct differences across populations. Integrations in the TN, B + , and B + R+ populations favored in gene localization (Fig. 2b; lower panel), which agrees with previous reports[7,45,46]. Notably, the proportion of genic integrations in the R+ population was significantly lower than that in the TN, B + , and B + R+ fractions. Integrations away from the centromere, as per the CIRCOS plot distribution (Fig. 2c), were similarly favored by all Timer fractions. HIV-1 integration specifically favors highly expressed gene bodies[47,48]. Therefore, HIV-1 provirus expression is usually studied in the context of the expression level of its host gene, which can have differential yet controversial effects according to the provirus insertion directionality[49,50]. We next obtained public RNA-seq datasets from parent Jurkat T cells to look into the basal expression level of the genes with integrated proviruses from our dataset. HIV-1 proviruses in the R+ population tended to integrate into genes with lower basal expression compared to other Timer populations (Supplementary Fig. 6a) and also tended to integrate at longer distances from the transcriptional start site (TSS) (Supplementary Fig. 6b); however, neither of these tendencies was statistically significant. Interestingly, we observed a tendency for opposite integration relative to the host gene in the R+ population (Supplementary Fig. 6c) and further traced the tendency of opposite integration across Timer fractions categorized by the basal level of host gene expression. The same direction of integration was preferred in TN and B+ fractions. A tendency for opposite integration began to appear in the B + R+ fraction in the lowest expression category and was the dominant tendency in the R+ fraction in all expression categories (Fig. 2d).

Next, we investigated whether the integrations in the R+ population were related to distinctive epigenetic features. We obtained published Jurkat uninfected chromatin immunoprecipitation (ChIP) seq datasets (see Materials and Methods for details) and analyzed the frequencies of provirus integration sites within 2 kilobases (kb) of several histone marks by comparing random integrations to B+ and R+ populations (Supplementary Fig. 6d; upper panel). The HIV-1 provirus in both populations favored integration in proximity to active histone marks, but not to repressive ones. This is consistent with previous reports on the integration of HIV-1 provirus[38]. Subsequently, we compared the preference for integration near each histone mark between the B+ and R+ populations by calculating an odds ratio. The results showed more frequent integration near the repressive

histone marks in the R+ population than in the B+ population (Fig. 2e, left panel). We further investigated whether this tendency could be reversed by comparing R+ to the non-expressing subset (TN) and found that TN proviruses showed preferential enrichment close to active histone marks compared to R+ proviruses. (Fig. 2e, right panel). Zinc Finger (ZNF) genes are reported to be associated with repressive chromatin marks in CD4+ memory T cells and to support long-term persistence for HIV-1 integrated proviruses[51]. Therefore, we sought to additionally verify the tendency of R+ proviruses to integrate close to repressive histone marks, by checking their frequency of integration in ZNF genes in comparison to other Timer populations. We found that approximately 5% of all genic integrations of the R+ population occurred in ZNF genes, a significant difference in comparison to any other Timer subset (Fig. 2f). Additionally, we observed a similar tendency for ZNF integration between the TN and B+ subsets; and therefore, examined their comparative histone enrichment statuses (Supplementary Fig. 6d; lower panel). Enrichment of integrations close to active or repressive histone marks did not show a clear difference between the two populations. These findings indicate unexpected similarities between the TN and B+ subsets, at least at the level of epigenetic control of provirus expression, in contrast to R+ proviruses, which seem to be more strongly influenced by epigenetic control. Collectively, these data suggest the existence of two latent populations, TN+ and R + , with different compositions and/or provirus expression regulatory mechanisms.

## Establishment and integration characterization of HIV-1 Timer clones

Bulk analysis of heterogenous infected cells, each of which has a unique integration environment, provides only averaged data contributed by all the complex factors governing the establishment and maintenance of latency. To elucidate the clonal heterogeneity of HIV-1 latency, we established HIV-Timer clones, each of which harbored a single copy of the complete provirus. We performed limiting dilution of Jurkat T and THP-1 cells infected with either HIV$_{Timer}$ or HIV$_{TNGFR}$ pseudotyped viruses at 3 dpi (Supplementary Fig. 7). The cloning protocol allows the enrichment of latently infected cells, as infected cells with high viral gene expression are eliminated by CPE. We obtained 17 and 9 Jurkat and THP-1 HIV-Timer clones, respectively. Subsequently, we performed DNA-capture sequencing as previously described in ref. 52, which enables comprehensive provirus characterization, including the whole provirus sequence, its respective integration site, and clonal abundance. ISs in established HIV-Timer clones were mostly localized in genic regions (96%) and were widely distributed across human chromosomes (Fig. 3a). Host genes carrying the HIV provirus in the HIV-Timer clones showed variable gene expression levels (Fig. 3b). Simultaneously, we analyzed the proviral expression status of each clone using Timer-FP. HIV-Timer clones showed wide intra- and inter-clonal heterogeneity in terms of provirus expression, as shown in some representative results (Fig. 3c; upper panel). Notably, we did not observe a significant correlation between the basal host gene and proviral expression levels (Fig. 3d). We further examined the induction of provirus expression by T cell stimulants or TNF-α (Fig. 3c; lower panel, Supplementary Fig. 8, and Table 1). We additionally checked the level of NGFR expression before and

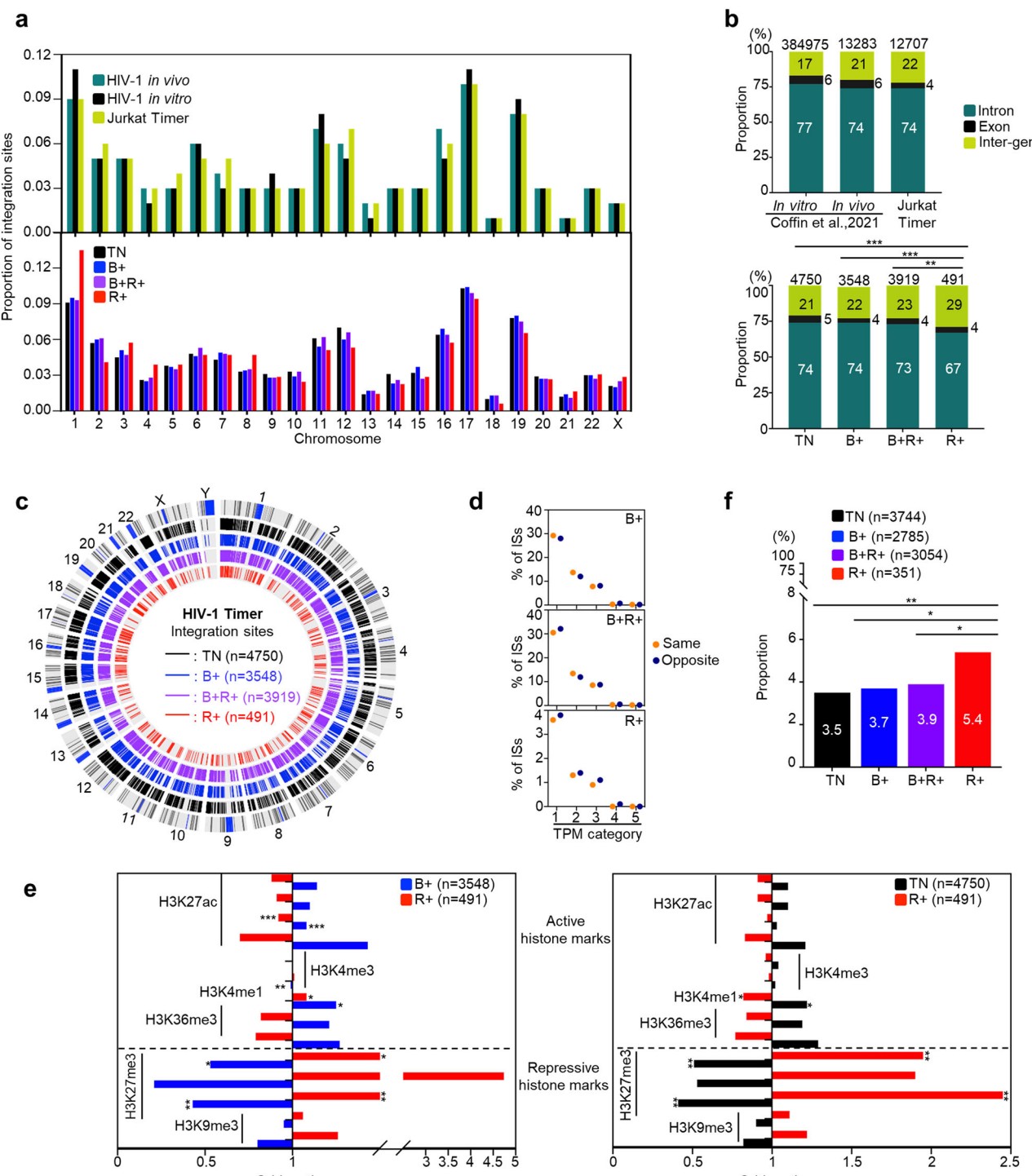

**Fig. 2 | HIV-1 integration landscape in HIV-Tocky in vitro system. a** Relative frequency of integration site distribution in each chromosome. Upper panel: data from previous study by Melamed A et al.[90] for HIV-1 in vivo and in vitro infections are plotted in teal and black respectively in comparison to data from the HIV-Tocky in vitro system (in lime-green). Lower panel: Comparing the frequency of integration sites across chromosomes in each sorted Timer population. **b** The frequency of unique ISs within genes or inter-gene distribution in the upper panel: in vitro infected PBMCs (2 donors), pre-ART HIV patients from previous study by Coffin J et al.[15] in comparison to total detected unique integration sites from Tocky in vitro system. Lower panel: The frequency of ISs within genes or inter-genes from different sorted Timer populations. The total number of integration sites analyzed in each fraction is shown at the top of each bar. **c** CIRCOS plot depicting viral ISs across the human genome in the HIV-Tocky in vitro system. Each chromosome is presented on the outer circle and is broken into sequential bins. Blue/gray, black, blue, violet,

and red bars indicate G-bands (condensed chromosome region by Giemsa staining for karyotyping), TN, B + , B + R + , and R+ populations, respectively. The number in the parathesis indicates the number of unique ISs detected. **d** Frequency of opposite or same integration across Timer fractions categorized by Transcripts Per Million (TPM); 1:0−50, 2:50−100, 3:100−500, 4:500−1000, and 5: > 1000. **e** The odds ratio of viral integration sites within ±2 kb of activating histone marks including H3K27ac, H3K4me3, H3Kme1, and H3K36me3, and the repressive histone marks with H3K27me3 and H3K9me3, comparing B+ and R+ Timer populations (Left panel), and comparing TN and R+ (Right panel). **f** The frequency of unique ISs within ZNF genes was calculated from total genic integrations in each Timer population, numbers in white on each bar denote the percentage of ZNF genes calculated from the total number of genic integrations shown as (n). For (**b, e, f**); Statistical significance was assessed by Fischer's exact test. * < 0.05, ** < 0.01, *** < 0.001.

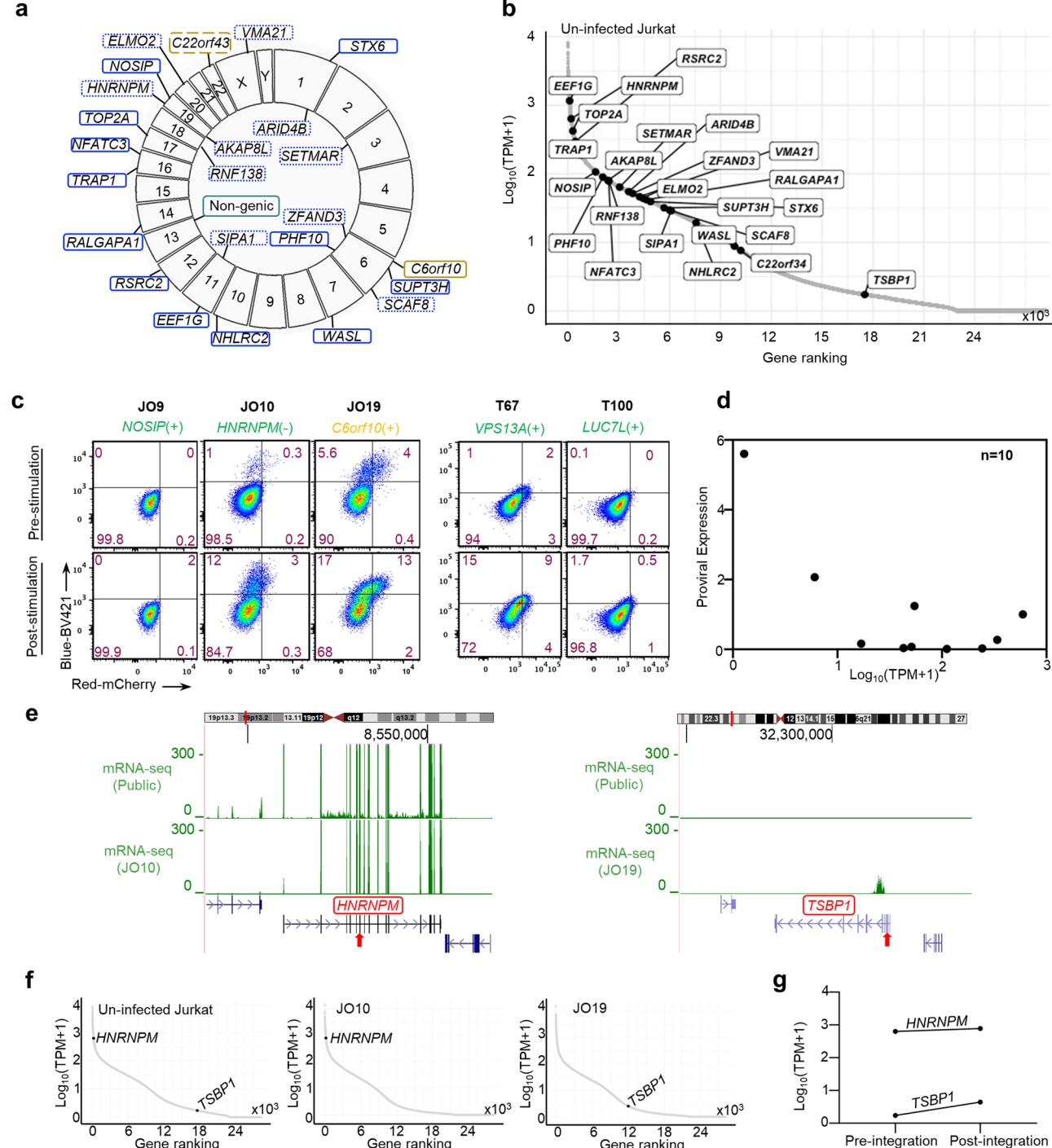

**Fig. 3 | Establishment and characterization of HIV-1 Timer clones. a** CIRCOS plot demonstrating the chromosomal integration site positioning for retrieved near full-length proviruses from a total of 17 Jurkat Timer clones. Color and line coding indicate genic/non-genic positioning, the orientation of integrated proviruses relative to the host gene, and gene transcription status in Jurkat cells. Active gene is squared in blue, silenced gene in gold. The solid line indicates the same orientation and the dotted line indicates the opposite orientation to the host gene. **b** Ranking plot for genes of integration of Timer clones. TPM values were obtained from RNA-seq data of uninfected Jurkat T-cell line (pre-integration data). **c** Flow cytometry plots showing Timer-FP expression denoting provirus expression in selected single integration Timer clones. Jurkat Timer clones' basal expression (upper left panel); provirus expression after T-cell stimulation for 24 h (lower left panel); THP-1 clones' basal expression (upper right panel) and THP-1 provirus expression after TNF-α

stimulation for 24 h (lower right panel). The host gene of HIV-1 integration in each clone is shown under the clone ID. ( + ) or (-) denotes the same or convergent orientation of provirus integration relative to the host gene, respectively. **d** Correlation scatter plot of the proviral expression percentages from single integration Jurkat Timer clones (n = 10) and the level of expression of the corresponding integration gene. **e** RNA-seq data of Jurkat uninfected T-cell line and RNA-seq data obtained from infected clones JO10 (right panel) and JO19 (left panel) are plotted. The position of the integrated provirus in each clone is indicated by a red arrow. **f** TPM values and ranking of each gene of integration of clone JO10 and JO19 are shown from RNA-seq data of uninfected Jurkat T-cell line (pre-integration, left panel) and after integration (middle panel; JO10) and (right panel; JO19). The observed change in TPM value pre- and post-integration within each clone is demonstrated in **g**.

**Table 1 | Characterization of the established HIV-1-Timer Jurkat clones**

| Serial number | Virus construct of origin | Clone ID | InGene | Provirus orientation to host gene | Basal proviral expression score | Cell-associated Viral DNA% | Provirus mutation | Provirus inducibility by T-cell stimulants | Transcription in Jurkat | A/B compartment in WT Jurkat cells (Hi-C) |
|---|---|---|---|---|---|---|---|---|---|---|
| 1 | HIV$_{Timer}$ | JO19 | *TSBP-1(C6orf10)* | Same | (++++) | 48 | pol: G539E vif: P49S | Inducible | Silenced | B |
| 2 | | JO23.5.1 | *C22orf34* | Opposite | (+++) | 48 | LTR: modulatory region (−127) | Inducible | | |
| 3 | | JO10 | *HNRNPM* | Opposite | (++) | 48.5 | no | Inducible | Active | A |
| 4 | | JO22 | *ARID4B* | Opposite | (++) | 53 | Pol: I2503K Vpu: R6203K | Inducible | | |
| 5 | HIV$_{TNGFR}$ | A6 | *TOP2A* | Same | (+) | 33 | Pol: P2725S | Non-inducible | | |
| 6 | | A8 | *NHLRC2* | Same | (+) | 64 | gag: G140E | Non-inducible | | |
| 7 | HIV$_{Timer}$ | JO9 | *NOSIP* | Same | (-) | 34 | vif: A152V | Non-inducible | | |
| 8 | HIV$_{TNGFR}$ | A2 | *RSRC2* | Same | (-) | 47 | no | Inducible | | |
| 9 | | A3 | *VMA21* | Opposite | (-) | 12.5 | LTR: modulatory region (-163) | Inducible | | |
| 10 | | A7 | *ZFAND3* | Opposite | (-) | 49 | no | Inducible | | |

after stimulation to rule out general activation of the cellular transcriptional status as a reason for increased provirus expression observed by Timer-FP after clones' stimulation (Supplementary Fig. 9). We observed a constant level of NGFR expression before and after stimulation. We next obtained available RNA-seq and ChIP-seq data for activating and repressing histone marks of parent Jurkat T cells to obtain a close-in view of the pre-integration environment in two representative examples of HIV-Timer clones: JO10 and JO19. Clone JO10 has a provirus integrated into an active transcribing gene (*HNRNPM*) (Fig. 3c and Supplementary Fig. 10a; left panel) while showing a relatively low basal Timer expression (Fig. 3c). In contrast, in clone JO19, we observed a high basal expression for a provirus integrated into a silenced gene (*TSBP1*) (Supplementary Fig. 10a; right panel). Pre-integration histone mark enrichment and HI-C compartment information[53] confirmed the basal transcriptional activity of the host gene in each clone (Supplementary Fig. 10a, b, and Table 1). We further investigated whether provirus integration altered the transcriptional activity of the host genes by performing RNA-seq analysis of these two HIV-Timer clones. We did not observe any significant change in the local transcriptome of the host gene after HIV integration in JO10 (Fig. 3e, left panel) or JO19 (Fig. 3e, right panel). Lastly, we quantified the transcription level using the transcripts per million (TPM) value and found a 3.5 and 1-fold increase in the transcriptional activity after provirus integration in clones JO19 and JO10 respectively (Fig. 3f, g). Notably, the proviruses were integrated in the opposite direction in JO10 and in the same direction as the host gene in JO19. Transcriptional interference (TI) is defined as the effect exerted by one transcriptional process on a second one in *cis*[54]. When the HIV-1 provirus genome is integrated into a host gene; polymerase complexes initiating at the host gene promoter will be continuously running into the integrated provirus. The fate of the integrated provirus transcription is, therefore, orientation-dependent as previously reported by ref. 49 which reported a dramatic decrease in the provirus transcription when integration is in an opposite orientation to the host gene owing to TI. In the current study, we observed a contradiction between the provirus expression and the basal gene activity status in clones JO10 and JO19. This contradiction can be partially explained by the TI phenomenon[49]; however, further investigation of the integration environment of each Timer fraction within a clone can provide valuable information on latency maintenance determinants. Another interesting observation was the non-inducibility of some clones that have their provirus integrated into an actively transcribing gene and in the same orientation as the host gene (Fig. 3c, Supplementary Fig. 11, and Table 1). Some studies have previously shown that HIV-1 promoter repression is associated with the CpG islands hypermethylation in in vitro T-cell models and determines the stability of HIV-1 latency[55,56]. We thought to examine the CpG methylation profile of the HIV-1 promoter by performing bisulfite sequencing for two non-inducible clones (JO9 and A8), and one inducible clone as a control (A3). We checked the methylation patterns in several CpG islands reported for HIV-1 5'LTR. JO9 showed 1.7-fold higher CpG methylation than the inducible clone A3 (Supplementary Fig. 12). However, Clone A8 showed the lowest methylation pattern. Our findings came in line with the previous reports in Jurkat cell lines suggesting that 5'LTR methylation is not required for HIV-1 promotor silencing[56,57]. Further analysis for more clones with and without various provirus expression-inducing agents could confirm DNA methylation as one of the possible reasons for the silencing and long-term survival of these clones. Collectively, these data demonstrate the successful establishment of a series of HIV-1 latent clones with heterogeneous provirus expression and variable integration into host genomic regions. Moreover, we were able to concomitantly characterize the whole proviral sequence and proviral expression for each clone, thereby offering a valuable resource for further investigation of HIV-1 latency mechanisms.

Ten single integration Jurkat Timer clones infected with HIV$_{Timer}$ (1-4, and 7) and HIV$_{TNGFR}$ (5,6, 8−10). Provirus integration and mutation information were obtained from DNA-capture seq results. Basal proviral expression score is assessed by flow cytometry % of Timer Blue expressing cells without any external stimulation; (-): 0, (+): 0-0.5, (++): 0.5-1.5, (++

+): 1.5–3 and (++++): > 3. PMA (20 ng/ml) and Ionomycin (1 μM) were used for T-cell stimulation. The inducibility of provirus was assessed 24 h later and shown in Fig. 3c and Supplementary Fig. 8. Cell-associated viral DNA was calculated by the following formula (copy number of HIV-1 gag DNA)/ [(copy number of Albumin)/2] x100. Transcription in Jurkat and chromatin compartment information were retrieved from parent Jurkat RNA-seq and Hi-C datasets (Supplementary Figs. 10 and 11).

## Application of HIV-Timer clones for the evaluation of Latency promoting agents

One possible application of HIV-Timer clones is in drug screening for LPAs or LRAs. Although several latent cell models harboring fluorescent proteins currently exist for monitoring proviral expression, HIV-Timer clones may have some advantages because of their unique characteristic of spontaneous transition from blue to red fluorescence upon provirus transcriptional arrest. Since the maturation half-life of Timer Blue to red chromophore is 4 h, we can detect reactivated or recently silenced proviruses by Timer fluorescence in a highly sensitive manner. Thus, we aimed to investigate the potential of the HIV-Tocky system to capture changes in the provirus transcriptional status when challenged with LPAs. First, we used J-Lat 10.6 cells to evaluate LPA activity as a representative of the conventional GFP model. In J-Lat 10.6 cells, a GFP coding sequence is inserted in the *nef* region to monitor proviral expression. Subsequently, we treated the cells with TNF-α to reactivate proviral transcription before adding the LPA reagent. Forty-eight hours after TNF-α stimulation, we treated cells with either an LPA or dimethyl sulfoxide (DMSO) for an additional 12 h (Fig. 4a). We used 2 classes of LPAs: Levosimendan, which promotes *Tat* degradation, and Triptolide, which inhibits NF-κB activity. We used three different drug concentrations according to previous reports[58,59] and performed a cell viability assay to exclude any non-specific cytotoxic effects on the result. No obvious toxicity was observed at all tested concentrations (Supplementary Fig. 13). We did not observe any change in GFP positivity during the initial 12 h after drug treatment (Fig. 4b,c, and Suppl. Fig. 14). Next, we treated the HIV-Timer clone JO19 with the same LPAs or DMSO using the same conditions as those for J-Lat 10.6 cells (Fig. 4a) and found a significant decrease of B+ cells and a significant increase of R+ cells (Fig. 4d, e) with both LPAs used in comparison to DMSO treatment. Timer transitions in response to Levosimendan or Triptolide are summarized in Fig. 4e. To validate Timer-FP specificity to capture the HIV-1 proviral silencing rather than being affected by the general cellular transcriptional shutdown, we quantified the levels of GAPDH mRNA in clone JO19 treated with LPAs for 12 h and compared them to samples treated with DMSO and TNF-α for 48 h as controls. GAPDH expression showed similar levels of expression in all the tested samples (Supplementary Fig. 15). Taken together, we demonstrated that the HIV-Tocky system can capture proviral transcription dynamics under latency-promoting effect in a highly sensitive manner compared with the conventional GFP latent infection model.

## Discussion

In vitro HIV-1 reporter systems have advanced the HIV-1 latency research in several respects. HIV-1 establishes its reservoir at an early stage of infection in vivo[21,60–63]. However, elucidating the mechanisms of latency establishment necessitates marking very rare latent infected cells[64] which in turn requires the use of several in vitro models that could recapitulate the situation in vivo[22,26,38,65]. Despite recent revolutionary advances in scrutinizing latent reservoirs ex vivo at the single-cell level[16,66–68], the most recent report by Sun et al. concluded the inability to identify a single universal marker for latently infected cells[69]. This signifies the need to improve HIV-1 reporter models to obtain guiding information regarding the obscure latent behavior of HIV-1. The development of the HIV-1-GFP reporter[70] and its use for cell line infection[26,71] have paved the way for later marking and separation of infected cells from purely latent cells in dual-reporter systems[23,24]. A GFP-modified reporter (HIV_GKO) facilitated better separation of latently infected cells from infected primary CD4+ T cells in vitro and provided additional evidence on the reactivation heterogeneity of the latent

population[25] reported previously[13,39]. Notably, all the aforementioned models used stable GFP—which has a long half-life (t ½ = ~56 h)—as a reporter[29]. This means that some expressing proviruses might lose their expression capacity at a certain point; however, this will not be reflected upon the status of GFP positivity because its decay time has not yet been reached. Pearson et al.[72] used a modified short half-life d2GFP (t ½ = ~3.6 h) to provide a more effective measurement of provirus transcription rates. Additionally, Venus fluorescence under LTR control[73] has been used to analyze the dynamic regulation of LTR activity. Both systems demonstrated provirus reactivation and silencing kinetics in a time-discernible manner; however, any temporal heterogeneity of the silencing or the stochastic re-expression can't be further tackled. Here, we propose and validate a novel HIV-1 reporter system for monitoring proviral expression dynamics with higher temporal resolution than conventional GFP systems by introducing a Timer-FP into a recombinant HIV-1-derived vector (HIV-Tocky) (Fig. 1a, b). In vivo, the risk of viral rebound is maintained by a rare minority of cells harboring intact latent proviruses[12,13]. In this study, we used single-round-infection-cycle HIV molecular constructs to limit the appearance of defective proviruses and precisely observe the post-integration dynamics of HIV-1 provirus. The HIV-Tocky system captured the dynamics of the very early phase of HIV-1 infection, including the simultaneous productive infection and latent reservoir establishment as reported in vivo[64], in a time-sensible manner in the primary CD4 + T cells, Jurkat T, and THP-1 myeloid cell lines (Fig. 1 c, g, and Supplementary Fig. 2).

In the HIV-Tocky system, the rare surviving cells that are expected to contribute to early reservoir seeding are represented by the scarce R+ fraction that survived and lost expression before joining the initially silent fraction (TN)[19,21]. To rule out the possibility that the monitored kinetics were affected by the selective growth advantage of the uninfected cells, we additionally constructed HIV_TNGFR with an expression cassette for NGFR, which enabled us to distinguish purely latent from uninfected cells. After the selection of infected cells, the same provirus dynamics as in the HIV_Timer were replicated (Fig. 1d). To the best of our knowledge, the current study is the first to successfully identify a latent cell subset with integrated proviruses that showed transient expression, which was subsequently terminated (R + ). Using bulk integration site analysis, we first verified the generation of heterogeneous clones in our HIV-Tocky in vitro model as previously reported in our WIPE assay model[22]. R+ proviruses demonstrated distinct integration features compared with other Timer populations. They showed a tendency to integrate more in non-genic regions and with orientations opposite to those of the host genes. Genic integration in ZNF-encoding genes was significantly higher in the R+ population. R+ proviruses were more likely integrated in close proximity to transcriptionally repressive chromatin regions. These findings are consistent with some reported attributes suggestive of "deep latency" in intact proviruses retrieved from elite controllers[74] or individuals with HIV under prolonged ART[75], which are shielded from host immune selection. However, we observed such features in a population of proviruses that lost their expression just a few hours after infection, apart from the long-term immune selection forces. HIV-1 latency at the transcriptional level is governed by proviral intactness, host cell factors, transcriptional interference, and epigenetic modifications. In contrast, stochastic provirus expression as mediated by the *Tat*-feedback loop was reported to control infected cell fate regardless of host cell status or chromatin organization[76,77]. In our dataset, as explained above, the criteria suggestive of "stable" latency were more significant in the R+ proviruses than in TN proviruses, which are proviruses with a decided lack of expression since the onset of infection. A striking finding was the genetic and epigenetic similarities existing between TN and B+ integrations. Given these observations, we can assume that the TN population has a heterogeneous composition of proviruses that are more readily able to stochastically turn on expression when aided by the *Tat* feedback loop or host cell factors. Conversely, R+ is a more controlled population with determined integration features into a quiescent destiny, guided by heterochromatin insertion or opposite directionality. Taken together, our data describe two different layers of latency with differential regulatory determinants at which

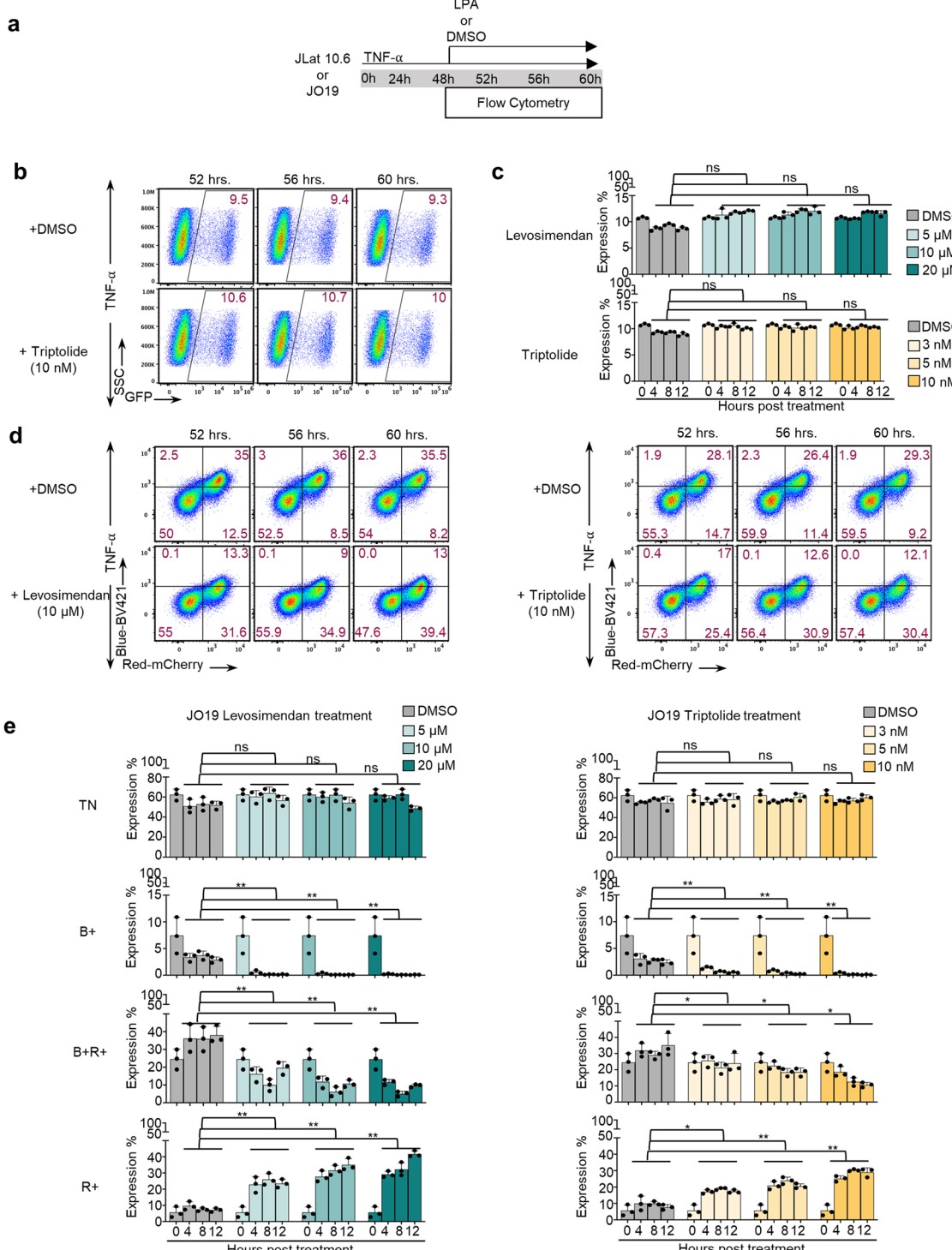

**Fig. 4 | HIV-1 Timer clones as a model for estimating LPAs efficacy.** LPAs effect on provirus silencing monitored by Timer transitions or GFP. **a** Assay overview. Schematic of experimental procedure involving drug treatment of Jurkat Timer clone JO19 or JLat 10.6. Briefly, JO19 or JLat 10.6 cells were stimulated with TNF-α (10 ng/μL) for 48 h, an LPA drug or DMSO was further added for 12 h, and the changes in Timer expression or GFP were monitored, respectively. **b** JLat 10.6 cells were stimulated with TNF-α (10 ng/μL) for 48 h, an LPA drug or DMSO was further added for 12 h, and the changes in GFP expression were monitored. Flow plots demonstrating GFP transitions under Triptolide (10 nM) treatment are shown. Flow plots representing Levosimendan (10 μM) treatment are shown in Supplementary Fig. 14. **c** Bar graphs representing the expression percentage of GFP are plotted against time under Levosimendan treatment (upper panel) or Triptolide (lower

panel) with 3 stated concentrations in comparison to DMSO treatment (*n* = 3 biologically independent experiments, mean ± SD). **d** Flow plots demonstrating Timer transitions over 12 h treatment with DMSO (upper left panel), Levosimendan (10 μM) (lower left), or DMSO (upper right), Triptolide (10 nM) (lower right panel). **e** Transitions of each Timer population from zero point until 12 h post-treatment are graphed by expression percentage obtained from flow cytometry analysis of each Timer population against time under Levosimendan treatment (left panel) or Triptolide (right panel) with 3 stated concentrations in comparison to DMSO treatment (*n* = 3 biologically independent experiments, mean ± SD). Statistical significance in (**c**) and (**e**) was assessed by Paired *t* test; ns= not significant, * < 0.05, ** < 0.01, *** < 0.001.

HIV-1 provirus reactivation and silencing appear to be different phenomena, with the former being more stochastic and the latter being more epigenetically controlled.

Next, we generated a library of Timer single-integration clones that could help in future investigations of latency determinants. Timer clones showed a wide variety of integration localities and basal target gene expression. In our model, in which *Tat* expression and basal LTR promoter activity were intimately coupled, the intra- and inter-clonal heterogeneity of provirus expression agreed with what has been previously monitored[41]. After ruling out cell environment differences in a uniform cell line, as well as the role of provirus mutations by confirming *Tat* and TAR intactness in our clones, we examined the pre-integration environment as a possible determinant of clonal heterogeneity. Our clones showed a preference to integrate into actively transcribed genes (eight out of ten) following the HIV-1 preferred integration pattern but in contrast to ref. 26, where they have enriched for latently infected cells before clone generation. A detailed mechanistic analysis of the expression pattern of each clone was not performed in the present study; however, the behavior of the Timer clones, despite their limited number, shares some features with previous in vivo or ex vivo reports. For instance, intact non-inducible proviruses[13] and active expression of persistent proviruses in ART-treated PLWH[78]. One notable finding was the high expression of the provirus in two out of ten clones (Table 1, Supplementary Fig. 10a; right panel), despite integration in the heterochromatin region. The factors governing latency reversal are complex. The dynamics of provirus transactivation by *Tat* have been thoroughly studied and proven to be stochastic[79]. The *Tat*-positive feedback loop shows a robust effect that could determine the fate of provirus expression regardless of cell-state changes[77,80]. Moreover, the *Tat* feedback loop allowed the induction of expression even in proviruses integrated near human endogenous retroviruses (HERV) LTRs[76], which are known to localize in heterochromatin regions[81]. Although this may partially explain the phenomenon, extended experiments by sorting and exploring the integration environment of each Timer population within each clone can provide more clues regarding the molecular mechanisms of provirus silencing and expression. Overall, the available set of the Timer clones demonstrated a reasonable variety of integration and provirus expression which therefore can be used as a novel tool for investigating the mechanisms of latency establishment and reversal aided by the high-resolution tracing of provirus kinetics by Timer fluorescence dynamics.

The ideal HIV treatment should result in remission and/or eradication. Assuming the uniformity of the HIV-1 latent reservoir, the "shock and kill" cure strategy aims to induce provirus expression to facilitate their immune clearance or virus cytopathy. "Block and lock" is an alternative strategy for achieving HIV-1 functional cure that entails enhancing the provirus latent state and getting the provirus promotor into deep and irreversible latency via epigenetic modifications, resulting in a drug-free non-remission status. In the current study, we demonstrated the feasibility of Timer fluorescence transitions to reliably capture the provirus silencing kinetics which was not possible by GFP (Fig. 4) using two LPAs with different modes of action on *Tat* transactivation. In either case, GFP dynamics remained stable, while the shut-off of provirus expression was clearly represented by the increasing percentage of the R+ population over the same drug exposure time in the Tocky model. Therefore, we demonstrated the ability of the Timer-FP to capture the rapid dynamics of silencing by targeting the Tat-LTR axis. Further experiments on epigenetic changes at the IS, especially in the R+ fraction, can elucidate more on the mechanisms of irreversible provirus silencing. In conclusion, we propose a novel reporter system for HIV-1 provirus expression dynamics that enables the analysis of the multilayered nature of HIV-1 latency by revealing real-time dynamics of provirus expression. Future studies with more detailed downstream analyses of these dynamic fractions can provide insights into the nature and determinants of their behavior. We propose the HIV-Tocky model as a useful tool for drug screening aided by high-throughput flow cytometry, particularly for modulators of HIV-1 expression fluctuations, which can enhance irreversible latency.

Additionally, we showed that the HIV-Tocky system recapitulated the expected dynamics in primary CD4 T-cells and cell lines of different lineages. Different cellular reservoirs deal differently with HIV-1 integration and latency. For example, in monocyte-derived macrophages, proviruses have less preference for integration in actively transcribing genes[82]. Future applications in primary myeloid and lymphoid cells may provide more information on latency dynamics and determinants in different tissues and anatomical reservoirs.

## Limitations of the study
This study has some limitations. Firstly, we utilized an in vitro model of provirus kinetics that lacked the immune selection effect and relied solely on the virus-induced CPE for clearance. Further analysis using in vivo models is required for more accurate visualization and assessment of HIV-1 provirus kinetics. Secondly, the THP-1 cell line was used for infection without differentiation into macrophage-like cells, and thus, it doesn't fully reflect all aspects of macrophage infection. Finally, based on the multiplicity of infection calculation in our experiments, some infected cells may contain more than one copy of the HIV-1 provirus, potentially introducing some bias in the interpretation of the integration data.

## Materials and methods
### Plasmid construction
We used two recombinant viruses, pNL4-3/ΔBglII/Timer (HIV$_{Timer}$) and pNL4-3/ΔBglII/Timer-EF1α-ΔNGFR (HIV$_{TNGFR}$). Both plasmids were derived from pNL4-3/ΔBglII[83]. The *env* gene was removed by BglII[84]. The *nef* gene between the *env* gene and the XhoI restriction site was replaced with the Timer or Timer-EF1α-ΔNGFR gene.

### Cell culture
HEK293T cells (human embryonic kidney cell line; American Type Culture Collection (ATCC)) were used for virus production. Jurkat T cells (E6.1 from ATCC) and THP-1 cells a human monocytic cell line; kindly provided by Dr. Hiroaki Takeuchi [Tokyo Medical and Dental University] were used for infection. JLat10.6 cells were obtained from the National Institutes of Health AIDs reagent Program. HEK293T cells were maintained in Gibco Dulbecco's Minimal Essential Medium (Gibco, Thermo Fisher Scientific), and suspension cell lines were cultured in Roswell Park Memorial Institute Medium (RPMI 1640; Gibco, Thermo Fischer Scientific). All culture media were supplemented with 10% heat-inactivated fetal bovine serum, penicillin (100 μ/ml), and streptomycin (100 μg/ml). All cell lines were maintained at 37 °C, 5% $CO_2$. Suspension cells were kept in T25 flasks and passaged twice a week, with a cell-to-fresh medium dilution ratio of 1:3.

### Virus production and infection
Pseudotyped HIV$_{Timer}$ and HIV$_{TNGFR}$ virus stocks were generated by transient co-transfection of HEK293T cells with a plasmid encoding either HIV$_{Timer}$ or HIV$_{TNGFR}$ and VSV-G using Lipofectamine 3000 transfection reagent (Invitrogen, Cat# L3000-015) according to the manufacturer's protocol. 293T cells were seeded in a 10 cm dish at a count of 1 million cells/dish. Fifteen μgs of the required plasmid were used per transfection experiment. At 28–40 h post-transfection, supernatants were harvested and filtered through a 0.45-μm-pore-size filter to clear cell debris. The virus was precipitated by centrifugation at 13,200 × *g* for 60 min at 4 °C. Concentrated virions were resuspended in complete RPMI 1640 medium, aliquoted, and stored at −80 °C. The amount of p24 Gag in the concentrated virus solution was quantified using HIV-1 p24 antigen enzyme-linked immunosorbent assay (ELISA) kit (ZeptoMetrix, Cat# 0801002) according to the manufacturer's instructions. Jurkat T and THP-1 cell lines' infection with HIV$_{Timer}$ or HIV$_{TNGFR}$ molecular constructs was performed by cells' incubation with the viral supernatant diluted in complete RPMI 1640 medium at 10 ng of p24 per $5 \times 10^5$ cells seeded in T25 flask. On assigned sampling time points (12 h until day 7), one-third medium change was performed. Unless otherwise specified, the final virus concentration used for infection was equivalent to 10 ng, as titrated by p24 ELISA. All HIV-1

infection and infected cell lines were handled in bio-containment level 3 rooms at Kumamoto University.

## Primary CD4+ T-cells isolation and infection

Primary CD4+ T-cells were extracted from healthy donors' PBMCs using density centrifugation on Ficoll-Paque Plus gradient (cytiva, Cat# 17144 002). Resting CD4+ T-cells were enriched by negative depletion in two successive steps, first labeling with Biotin anti-human CD8 a (BioLegend, Cat # 301004) and Biotin-TCR γ/δ Antibody (BioLegend, Cat#331206) negative and secondly adding Pan T cell Isolation cocktail (Miltenyi Biotec, Cat#17144002) according to manufacturer instructions. Isolated cells were then cultured in RPMI with 10% fetal bovine serum, penicillin (100 μ/ml), and streptomycin (100 μg/ml) with additional recombinant human IL-2 (PEPROTECH, Cat#200-02) at a final concentration of 100 μ/ml. Next, cells were activated with Dynabeads Human T-Activator αCD3/CD28 activating beads (gibco by Life Technologies, Cat# 11131D) at a concentration of 1 bead/cell in the presence of IL-2 for two days. We infected primary CD4+ T-cells using the RetroNectin-Bound Virus infection method (TaKaRa, Cat#T100A/B) according to the manufacturer's instructions. $HIV_{Timer}$ construct was used for infection at a concentration of 500 ng of p24 per $1 \times 10^6$ cells. Cells were collected at 1 dpi until 6 dpi and analyzed by flow cytometry.

## Timer flow cytometric analysis and cell sorting

For integration site analysis, Jurkat T cells were infected with the $HIV_{Timer}$ virus construct as described above. Infected cells ($5 \times 10^5$) were aliquoted at 12 h, and 1–7 dpi and cryopreserved. Time-course flow cytometric analysis was performed by thawing and analyzing all time points in the same setting. For Timer-only fluorescence detection (For infected primary CD4 + T-cells and Jurkat T-cells), cells were revived in pre-warmed complete RPMI 1640 medium, washed once with PBS, and incubated in near-IR fluorescent live/dead fixable dye (Thermo Fischer, Cat# L34972) to stain dead cells at a 1:1000 dilution in PBS for 30 min at 4 °C. Cells were then washed once and then fixed with 1% PFA in PBS for 15 min at 4 °C in the dark before data acquisition. To determine the surface NGFR expression along with Timer, revived cells were first stained for dead cells as described above, and after a single wash in PBS, they were stained with APC-conjugated anti-human CD271 (NGFR) antibody (BioLegend, Cat# 345108) at 1:30 final concentration for 30 min at 4 °C. Cells were washed twice with PBS and fixed in 1% PFA in PBS for 15 min in dark at 4 °C before data acquisition. Timer and ΔNGFR fluorescence were measured and/or sorted using SH800 sorter (Sony). Timer Blue fluorescence was detected in the FL1 channel (450/50 nm) excited by a 405 nm laser, while the red fluorescence was detected in the FL3 channel (617/30) excited by a 561 nm laser. Data were analyzed using FlowJo 10.7.1 software (Tree Star, Inc.). For viral mRNA quantification, Jurkat T cells were infected with the $HIV_{TNGFR}$ virus construct as described above. The infected cells were collected from day one to 7 post-infection and cryopreserved. Aliquots were then resuspended in warm complete RPMI 1640 medium, washed once with PBS, and stained with near-IR fluorescent live/dead fixable dye (Thermo Fischer, Cat# L34972), followed by staining with APC-conjugated anti-human CD271 (NGFR) antibody (BioLegend, Cat # 345108) at 1:30 final concentration as described above. Cells were finally washed twice with PBS and resuspended in 1% FCS in PBS buffer. The Timer populations were sorted using BD FACSAria III machine at the biocontainment level 3 facility at Kumamoto University.

## qRT PCR for HIV and NGFR transcripts quantification

RNA (Fig. 1h and Supplementary Fig. 3b) was extracted by RNeasy Mini Kit (Qiagen, Valencia, CA, USA) according to the manufacturer's instructions with DNaseI treatment. cDNA was synthesized using ReverTra Ace® qPCR RT Master Mix (Toyobo, Osaka, Japan, Cat#FSQ-201) according to the manufacturer's instructions. Relative cellular HIV mRNA levels were quantified using SYBR qPCR assay using primers to detect unspliced *Gag* mRNA and multiple spliced *tat/rev* transcripts as previously described[85]; 5' for- TCA GAC TCA TCA AGC TTC TCT ATC AAA GC-3' and 5' rev-

GAT CTG TCT CTG TCT CTC TCT CCA CC-3.' Viral transcripts were normalized to 18S rRNA. To detect NGFR mRNA, we used the following primer set: Forward 5'-ACA GGC CTG TAC ACA CAC AG-3'and Reverse: 5'-CGT GCT GGC TAT GAG GTC TT-3'. NGFR transcripts were normalized to GAPDH. The assay was performed using an Applied Biosystems StepOnePlus Real-Time PCR System (Thermo Fisher Scientific). Relative cell-associated HIV mRNA copy numbers were determined in a reaction volume of 20 μL with 10 μL of Thunderbird SYBR qPCR mix (Toyobo, Osaka, Japan Cat#QPX-201T), 0.3 μM of each primer and 2 μL of cDNA. Cycling conditions were 95 °C for 1 min, then 40 cycles of 95 °C for 15s and 60 °C for 1 min. Real-time PCR was performed in triplicate and relative cell-associated HIV mRNA copy number was normalized to cell equivalents using 18S rRNA or GAPDH expression and the comparative Ct method[86].

## Digital droplet PCR

Droplet digital PCR (ddPCR) for quantifying cell-associated viral DNA was performed by using primers and a probe targeting the HIV-1 *Gag* region and the ALB gene, according to the previous report with minor modifications[52]. ddPCR droplets were generated by the QX200 droplet generator (Bio-Rad). Generated droplets were then transferred to a 96-well PCR plate and sealed with a pre-heated PX1™ PCR plate sealer (Bio-Rad) for 5 s at 180 °C. For probe-based ddPCR, PCR was performed with the following settings: 95 °C for 10 min followed by 39 cycles of 94 °C for 30 s, 58 °C for 60 s, and final 98 °C for 10 min and 4 °C for hold. The plate was then placed in the QX200 droplet reader (Bio-Rad) for quantification of the number of positive and negative droplets based on their fluorescence. Data were analyzed using QuantaSoft software (Bio-Rad). Then DNA load was calculated as follows: Cell-associated viral DNA (%) = (copy number of HIV-1 *Gag* DNA)/((copy number of ALB)/2) × 100. Quantification of 2-LTR HIV DNA was performed according to the previous report[87].

## Amplification of near-full-length single HIV-1 genome

Nearly full-length single HIV-1 genome PCR was performed as described previously[22,35] with minor modifications. Briefly, genomic DNA was diluted to the single-genome level based on ddPCR and Poisson distribution statistics. The resulting single genome was amplified using Takara Ex Taq hot start version (first-round amplification). PCR conditions for first-round amplification consisted of 95 °C for 2 min; followed by 5 cycles of 95 °C for 10 s, 66 °C for 10 s, and 68 °C for 7 min; 5 cycles of 95 °C for 10 s, 63 °C for 10 s, and 68 °C for 7 min; 5 cycles of 95 °C for 10 s, 61 °C for 10 s, and 68 °C for 7 min; 15 cycles of 95 °C for 10 s, 58 °C for 10 s, and 68 °C for 7 min; and finally, 68 °C for 5 min. First-round PCR products were diluted 1:50 in PCR-grade water and 5 mL of the diluted mixture was subjected to second-round amplification. PCR conditions for the second-round amplification were as follows, 95 °C for 2 min; followed by 8 cycles of 95 °C for 10 s, 68 °C for 10 s, and 68 °C for 7 min; 12 cycles of 95 °C for 10 s, 65 °C for 10 s, and 68 °C for 7 min; and finally, 68 °C for 5 min. PCR products were then visualized by electrophoresis on a 1% agarose gel. Based on Poisson distribution, samples with ≤30% positive reactions were considered to contain a single HIV-1 genome.

## Establishment of Timer clones

For Jurkat Timer clones, cells infected with $HIV_{Timer}$ at 3 dpi were cloned by limiting dilution by seeding 0.3 cells per well in 96 cell culture plates. After approximately 3 weeks, 24 clones were obtained. We confirmed 11 clones for infection by qPCR for the *Gag* region (Supplementary Fig. 16). THP-1 clones were infected as described above in the virus production and infection subsection. To select the infected cells, NGFR MACS separation was performed according to the manufacturer's instructions. First, the infected cells were stained with a biotinylated anti-human CD271 (NGFR) antibody (BioLegend, Cat# 345122). Subsequently, the cells were magnetically labeled with Streptavidin MicroBeads (Miltenyi Biotec, Cat# 130-048-102). The cell suspension was then loaded onto a MACS column, which was placed in the magnetic field of a MACS separator. Magnetically labeled NGFR+ cells were retained in the column and further eluted as the positively selected cell

fraction after magnetic removal. NGFR+ cells were subjected to limiting dilution, as described for Jurkat clones. Twelve THP-1 clones were obtained. Additional 6 Jurkat Timer clones were generated using the NGFR+ selection method after infection.

## Bisulfite Cytosine Methylation analysis

Bisulfite sequencing was performed according to the previous report[56] with some modifications. Total genomic DNA was extracted using the DNeasy Blood and Tissue Kit (QIAGEN) according to the manufacturer's recommendations. Methylation analysis has been performed using Bisulfite conversion kit (Active Motif, Cat#55016). Bisulfite-treated DNA was amplified using PCR in a 50-μl reaction mixture using EpiTaq HS polymerase (TAKARA, R110A). The master mix contained 50 mM Tris-HCl (pH 9.2), 1.75 mM MgCl$_2$, each dNTP at 350 μM, and each primer at 45 pmol. Primers for the HIV 5′ LTR were used according to the previous report[56]. PCR was performed with about 50 ng of genomic DNA for 40 cycles at 95 °C for 60 s, 58 °C for 120 s, and 72 °C for 60 s. Amplification products were cloned in the pGEM-T-Easy Vector System (Promega, Madison, WI). Cloned DNA from positive clones was checked for specific bands and at least 10 positive bands from each Timer clone were sent for Sanger sequencing. Retrieved sequences were then trimmed to remove the bad quality scored reads at an Error Probability Limit of 0.05, aligned to the 5′LTR (HXB2) reference genome. Converted and un-converted Cytosine residues were counted and a percentage for total mCpGs was calculated relative to the total number of CpG islands checked using Geneiuos Prime 2023.2.1. (Supplementary Fig. 12).

## Timer and JLat clones' drug treatment

To check provirus inducibility in Timer clones, Jurkat Timer clones were treated for 24 h with a combination of T-cell stimulants: phorbol 12- myristate 13-acetate (SIGMA, Cat# P1585) at a final concentration of 20 ng/ml and ionomycin (Nacalai Tesque, Cat# 19444-91) at 1 μM final concentration. For THP-1 Timer clones, recombinant human TNF-alpha (hTNF-α, PEPROTECH, Cat# 300-01 A) at 10 ng/ml was used for 24 h. In latency-promoting experiments (Fig. 4), Timer or JLat 10.6 clones were seeded at $5 \times 10^4$ cells per 12 well plate and treated with hTNF-α at 10 ng/ml concentration. At 48 h, LPA or DMSO was added, and the cells were aliquoted at 4, 8, or 12 h for flow cytometric analysis. The following LPAs were used: Levosimendan (TGI, Cat# L0320) at 5, 10, or 20 μM final concentration and Triptolide (Cayman, Cat# 11973-1MG) at 3, 5, or 10 nM final concentration. Cell viability was assessed using cell counting kit-8 assay (Cat# GK10001). The JO19 cells were serially diluted and seeded according to the manufacturer's instructions. Treatment with LPA or DMSO was continued for 8 or 24 h before the addition of cell-counting kit-8 reagent. After 4 h of incubation, color absorbance from each well was measured using SPEC-TRAmax 190 microplate spectrophotometer (Molecular Devices, USA) using a 405 nm filter. The obtained absorbance was further analyzed using SoftMAx Pro Microplate analysis software (version 5, Molecular Devices, California, USA). Absorbance was normalized to the no-cell control, and the final cell survival percentage was divided by the DMSO control survival percentage and plotted (Supplementary Fig. 13).

## Linker-mediated (LM)-PCR and integration site data analysis

HIV-1 IS analysis was performed using LM-PCR and high-throughput sequencing as described previously in ref. 44. In brief, Timer populations from bulk-infected Jurkat cells with the HIV$_{Timer}$ molecular construct were sorted and genomic DNA was extracted using the DNeasy Blood and Tissue Kit (QIAGEN) according to the manufacturer's recommendations. About 2 μg of genomic DNA was sheared by sonication with a Picoruptor (Diagenode, S.A., Belgium) instrument to a size of 300–400 bp in length. DNA end was repaired and the addition of adenosine at the 3′ end of the DNA was performed with NEBNext Ultra II End Repair/dA-Tailing Module (New England Biolabs, Cat# E7546). A linker was ligated to the ends of DNA using the NEBNext Ultra II Ligation Module (New England Biolabs, Cat# E7595). The junction between the 3′LTR of HIV-1 and host

genomic DNA was amplified using primers targeting the 3'LTR and linker regions. The first PCR amplicons were purified using a QIAquick PCR purification kit (QIAGEN), and then the second PCR was performed. The following thermal cycler conditions were used for both PCRs: 96 °C for 30 s (1 cycle); 94 °C for 5 s, 72 °C for 1 min (7 cycles); 94 °C for 5 s, 68 °C for 1 min (13 cycles); 68 °C 9 min (1 cycle) and hold at 4 °C. The second PCR amplicons were purified using a QIAquick PCR Purification Kit (QIAGEN), followed by Axy Prep Mag PCR cleanup bead purification (AXYGEN, Cat# MAG-PCR-CL-50). Purified PCR amplicons were quantified using the Agilent 4150 TapeStation system, and quantitative PCR was performed using primers P5 and P7 (GenNext NGS library quantification kit, Toyobo, Code# NLQ-101). LM-PCR libraries were sequenced using Illumina MiSeq paired-end reads, and the resulting FASTQ files were used for downstream analysis. Three FASTQ files, including Read1, Read2, and Index Read, were obtained from the IIIumina MiSeq. Read1 corresponded to sequencing data generated by a primer within the HIV-1 LTR region and Read2 to data generated by a primer within the linker. The Index Read corresponds to the 8-bp index sequence in the linker. First, we identified clusters on the flow cell with high sequencing quality of the Index Read (Phred quality score >30 at each position of the 8-bp index read) using an R code kindly provided by Michi Miura (Imperial College London, UK). Subsequently, we removed adaptor sequences from Read1 and Read2 selected reads with the LTR sequence in Read1 (TAGCA for HIV-1) and performed a cleaning step to remove reads that were too short or had a too low Phred score. The cleaned sequencing reads were aligned to the Homo sapiens genome assembly hg19 with HIV-1 (GenBank: K03455.1) using the BWA-MEM algorithm[88]. We further processed the data to remove the unmapped reads, paired reads mapped to different chromosomes, and reads including 5' LTR sequences. We then exported files containing information on the integration sites, DNA shear sites, and the number of reads. BED files were generated from exported files containing information on integration sites. RefSeq gene data were obtained from UCSC tables (https://genome-asia.ucsc.edu/). We analyzed the genome integration site preferences. The positions of the RefSeq genes were compared to the integration sites using the R package hiAnnotator (http://github.com/malnirav/hiAnnotator).

## DNA-capture-seq library and data analysis

HIV-1 DNA-capture-seq was performed as described previously[52] with minor modifications. Briefly, gDNA was extracted from Timer clones using the DNeasy Blood and Tissue Kit (Qiagen) according to the manufacturer's instructions. 1 μg of genomic DNA was sheared by sonication using a Picorupter (Diagenode s.a., Liege, Belgium) to obtain fragments with an average size of 300 bp. Libraries for NGS were prepared using the NEBNext Ultra II DNA Library Prep Kit for Illumina (Cat# E7645) following the manufacturer's instructions. To enrich the viral fragments contained in the synthesized libraries, several libraries were pooled to perform the capture step. The pooled libraries were mixed with virus-specific 161 biotinylated probes in the presence of human Cot-1 DNA (Invitrogen, Cat#15279011) and xGen Universal Blocking Oligos (IDT) for the hybridization step. A series of washing steps were performed using DNA xGen lockdown reagents (IDT), following the manufacturer's recommendations. The quality of the enriched DNA libraries was evaluated and quantified by electrophoresis using the TapeStation 4150 system (Agilent Technologies). Finally, the multiplexed libraries were subjected to cluster generation using a MiSeq Reagent Kit v3 (150 cycles) in MiSeq sequencing systems (Illumina) with $2 \times 75$ bp reads. Three FASTQ files, Read1, Read2, and Index Read, were obtained from the Illumina MiSeq. We first performed a data-cleaning step using an in-house Perl script (kindly provided by Dr. Michi Miura, Imperial College London), which extracts reads with high Index Read sequencing quality (Phred score >20 at each position of the 8-bp index read). Subsequently, we removed adaptor sequences from Read1 and Read2 followed by a cleaning step to remove reads with Phred scores that were too short or too low, as previously described[44]. Cleaned sequencing reads were aligned to the

reference genome using the BWA-MEM algorithm[88]. To determine the complete sequence of the provirus in different cell lines and to infer its structure, the reference genome to which they were aligned included the entire human genome (hg19) and the complete HXB2 sequence as an independent chromosome. To properly determine the integration sites, the HXB2 sequence in the reference genome was included on two different chromosomes: viral LTRs (HIV_LTR) and proviral sequences without LTR (HIV_noLTR). We used SAMtools and Picard (https://github.com/broadinstitute/picard) for further data processing and clean-up, including the removal of reads with multiple alignments and duplicated reads. The final aligned files were visualized using Integrative Genomics Viewer (IGV 2.8.13).

### RNA-seq
For Jurkat Timer clones JO10 and JO19, RNA was extracted using the RNeasy Mini Kit (Qiagen) according to the manufacturer's instructions with DNase I treatment. We ensured RNA quality using the Agilent Bioanalyzer 4150 TapeStation system. An RNA library was prepared using the NEBNext Ultra II directional RNA library prep kit for Illumina and sequenced using Illumina NextSeq 500 to obtain single-end reads. Around 40 million reads were obtained using the following read length: read1:75 bp. RNA sequences were processed using Cutadapt, PRINSEQ, STAR alignment, and SAMtools. Three independent biological replicates were analyzed for each clone. The Jurkat mRNA-seq dataset was obtained from the NCBI Sequence Read Archive (SRA) database under the GEO accession number GSM2171783 and processed in the same manner. The final aligned files were visualized using the UCSC Genome Browser (https://genome-asia.ucsc.edu).

### ChIP seq datasets, epigenetic data analysis, and Hi-C
Chromatin data (ChIP-seq) from Jurkat uninfected T-cells was downloaded from Sequence Read Archive (SRA) data repository: H3K27ac (SRR12884765, SRR1603650, SRR1509753, SRR2043614), H3K4me3 (SRR577482, SRR577483), H3K4me1 (SRR7782877), H3K36me3 (SRR11783979, SRR11783980), H3K27me3 (SRR12884766, SRR647929, SRR11903008) and H3K9me3 (SRR13191912, SRR12884767). The FASTA files obtained were processed with in-house scripts using Cutadapt, PRINSEQ, BWA alignment, and SAMtools to obtain the mapped BAM files. Peak calling was performed using MACS2 and the corresponding input files for each histone mark. Features of histone modifications in the genomic environment flanking HIV-1 were analyzed using a Perl script (v5.22.0) as described previously[89]. Hi-C for parent Jurkat T cells was obtained from NCBI GEO (GSE122958). Hi-C correlation matrices were generated using FAN-C (https://github.com/vaquerizaslab/fanc) and Cooler (https://github.com/mirnylab/cooler), at a resolution of 1 Mb.

### Statistics and reproducibility
Statistical analyses were performed using GraphPad Prism 9 software (GraphPad Software, Inc., CA, USA). For Fig. 2b, e, f; Fischer's exact test was used to compare integration tendencies across Timer populations. Statistical significance was defined as: $*p < 0.05$, $**p < 0.01$, $***p < 0.001$. Data presented in Fig. 4c, e are obtained from three biologically independent experiments. Statistical significance was assessed by Paired $t$ test; ns = not significant, $*p < 0.05$, $**p < 0.01$, $***p < 0.001$.

### Reporting summary
Further information on research design is available in the Nature Portfolio Reporting Summary linked to this article.

### Data availability
Raw sequence data included in this study (Integration site, DNA capture seq, RNA-seq) have been deposited to SRA under the Bioproject: PRJNA1073283. Integration site analysis data for TN, B + , B + R + , and R+ can be found under the accession numbers: SRX23531115, SRX23531116, SRX23531117, and SRX23531118 respectively. DNA capture sequencing data for Timer clones: JO9, JO10, JO19, JO22, JO23, A2, A3, A6, A7, and A8 can be found under the accession numbers from: SRX23533209 to SRX23533218 respectively. RNA seq data for Timer clones JO10 and JO19 are found under accession numbers: SRX23545644 and SRX23545645 respectively. Bisulfite sequencing data have been deposited to NCBI GenBank under the accession numbers from: PP347485 to PP347542. The source data behind the graphs in the manuscript are shown in the Supplementary Data file. Any other data supporting the findings of this study are available from the corresponding author upon reasonable request.

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

## Acknowledgements

We are grateful to Y. Matsuoka, A. Murakawa, N. Monde, and H. Terasawa for their technical and administrative support. This work was supported by research grants from the Japan Agency for Medical Research and Development (AMED) (JP23fk0410052, JP23wm0325068, JP23jm0210074, and JP23fk0410040) awarded to Y.S., JP23fk0410057 to K.S., a Biotechnology and Biological Sciences Research Council (BBSRC) David Phillips Fellowship (BB/J013951/2), a Medical Research Council (MRC) grant (MR/S000208/1) to M.O.This work was supported by the JSPS Core-to-Core Program and JST MIRAI. The funders had no role in the study design, data collection, data interpretation, or discussions regarding submission for publication.

## Author contributions

M.O. and Y.S. contributed to the conceptualization of the study. O.R. and Y.S. designed the research; O.R., K. Monde, K.S., A.R., W.S., S.A.R. S.N.S, C.M., and H.T. performed the research; K. Monde, M.O., and H.T. contributed new reagents/analytic tools; O.R., K.S., B.J.Y.T., K.N., K. Maeda, and Y.S. analyzed the data; O.R. and Y.S. wrote the manuscript. All authors read the manuscript draft and approved the final version for submission.

## Competing interests

The authors declare no competing interests.
