## [Peer Review File · Communications Biology]

Reviewers' comments:

Reviewer #1 (Remarks to the Author):

In this manuscript, Reda et al described the development of a new reporter model to study HIV latency that uses a fluorescent Timer protein that changes its emission spectrum over time. They use this new reporter in two cell lines Jurkat and ThP1 and evaluate expression dynamics, provirus intactness, integration landscape and latency promoting agents. This new reporter system adds to the tools of other reporters previously used to study latency and add the advantage of the possibility to evaluate latency in cells that revert to a latent state after viral expression, which was impossible till now. In spite of this, there are several concerns with the manuscript as presented that will need to be addressed to have a meaningful impact on the field of HIV.

Major concerns.

- As stated in the limitations section, the reporter system is only validated in cell lines. It will be important to address whether this reporter system can be use in more relevant primary CD4 T cells and/or macrophages. Otherwise, the title should be change to "HIV-Tocky system to visualize proviral expression dynamics in cell lines"

- Figure 1. The MOI for the NGFR virus is above 1 and that can lead to multiple infections per cell as shown in sup Fig 2C. This could lead to misinterpretation of some of the results, in particular with bulk analysis. It is also not clear whether all Blue/Red cells are NGFR positive. It will be important to demonstrate that in each population generated (TN, B, R, BR) contains intact proviruses as it is well known that even with replication deficient HIV constructs, deletions can occur. In particular when other promoters and fluorescent proteins are introduced. A transcript control (like NGFR) should be included in parallel to Fig 1I to demonstrate that only the HIV promoter is silenced but not the NFGR. Also, it is not clear whether the analysis was done in Jurkat or ThP1

- In figure 2S, the 2-LTR data should also be shown and not as data not shown if it has been done. Even it is a replication deficient provirus, 2LTR and 1LTR can still form. Furthermore, the HIV DNA analysis performed does not distinguished between integrated and non-integrated HIV DNA.

- Figure 2. Because of the high multiplicity of infection, the integrating analysis in the 4 populations may be misleading as multiple integrations may be in an individual cell. This should be either address experimentally or mention as another caveat of the study.

- Figure 3. It is unclear which clones are from HIVTimer or HIVTNGFR. It should be demonstrated that each one has 1 copy per cell and that the provirus is intact. It will be beneficial to include the integration site for each clone in the figure C for clearness. Changes in NGFR expression concomitant with activation should be included to evaluate that the increased in blue/red is associated with changes in HIV transcription and not general transcription.

- Figure 4. The use of LPAs is interesting but it will be important to address if the reduction on transcription is just associated with HIV transcription or a shutdown of all transcription. For example, what are the levels of NGFR or other cellular protein? Also, Figure E is missing any statistical analysis.

Minor concerns

- Line 381 is missing a part of the sentence

Reviewer #2 (Remarks to the Author):

In this manuscript Reda et al. describe a new system to track HIV proviral integration and subsequent establishment of latency.

This ingenious system uses a timer protein that switches from "blue" to "red" fluorescence over time. The manuscript is well detailed, relevant for the HIV latency/cure field, and well articulated. The authors describe openly known limitations which is great for scientific rigour.

Major Comments:

-(Lines 198-199) It is not clear in Fig S4B that there are differences in distances from the TSS. The distances seem to distribute evenly, with different count number.

- (Lines 224-225) Although statistically significant, which is easy to achieve with the high number of replicates, it is not clear that the R+ integration in ZNF genes is biologically relevant, the authors could perhaps use other controls in the figure (one showing little integration, and another more abundant. It is hard to estimate and normalize "ZNF genes" as they are clustered as a family vs others, as their abundance is not stated.

- (Lines 270-271). The authors state that their clones JO#10 and JO#19 results contradict the interference phenomenon. (i) authors should further introduce the phenomenon for a wider audience (ii) it is perhaps strong to arrive such conclusions with only two clones. It could perhaps be achieved with 10+ clones.

-In Fig 3C, The JO#19 clone shows an increase of R+B+ cells vs JO#10, wouldn't that mean that it is somewhat independent of the TNF α stimulation? Shouldn't the timing of TIMER R+ be the same for both clones (with normalization to the 4% positive before stimulation?)

- The formation of defective proviruses should not be dependent on Env as the authors state. Although not clear what drives the high number of defective proviruses in vivo, it is likely a selection via CD8/NK cells that kills and cleans any cells expressing HIV (including full proviruses). The sentence should be revised.

Minor Comments:

- Line 85 the word spontaneously should be more characterized, later the authors describe the time it takes for Timer protein to change its spectrum. The text would benefit from a better description in this section.
- It is not clear to me what are the differences between the data of Figure 1C/D and Figure S1, please clarify in the text.
- The authors describe that they sorted different populations for B+, B+R+ and R+ in lines 150 to 154. It should be clear in the text the time post-infection where that was performed.
- Authors state that no 2-LTR circles were detected with data not shown. Such data (and proper controls, should be provided). Although at later time points 2LTR circles tend to disappear, explaining such observation it is important for the scientific rigor to present such data.
- (Line 370) Please clarify the usage of the word asymmetric, do the authors mean distinct/different/independent?
- The authors compare the "shock and kill" and "lock and block" strategies, describing a higher viability for the "lock and block" strategy. Wouldn't the transcription stochasticity (very well described in the manuscript), apply in a "lock and block" strategy as well?
- The THP-1 model should be described in the limitations of the study, especially because the authors did not differentiate the cells into "macrophage like" cells with PMA or other methods. Monocytes are not a "canonical" target of HIV-1, as they do not have the CD4 receptor.
- Please provide amounts of DNA, cell number, and plate type/area for virus production and infection.
- Please show Data referred to in line 529.

Reviewer #3 (Remarks to the Author):

Reda et al. developed a novel HIV-Tocky system which can identify active versus inactive HIV gene expression dynamics by measuring dual fluorescence emission spectrum shifting (Timer protein). By using this HIV-Tocky system, the authors applied interrogated HIV integration heterogeneity, clonal expressions, and tested latency stability under latency promoting agent (LTA) treatments. The HIV-Tocky system is elegantly designed to capture temporal dynamic changes after HIV reporter virus infection and the fan-shape changes of the Timer fluorescence proteins reflected the latent proviruses. The authors performed HIV integration sites analysis of active (B+) versus inactive (R+). In Figure 1, the authors showed the dynamic changes of Timer protein fluorophore over time in two cell lines, Jurkat and THP-1. In Figure 2, the authors examined the HIV integration site landscape of active (B+) versus inactive (R+) HIV infection. In Figure 3, the authors established long term culture of these HIV-timer clones. Among these 17 clones, 2 (11%) have HIV integrated into silenced genes while the remaining 15 have HIV integrated into actively transcribed genes. In Figure 4, The authors used this HIV-timer system to evaluate drug effect as a tool for potential clinical implications for drug testing.

The major strength of this manuscript is the elegant use of timer protein and the rigorous analysis of integration site and near full-length proviral genome sequencing. The striking finding is that inactive HIV

infection (R+) seems to have more integration into intergenic regions associated with repressive chromatin marks and integration into zinc finger genes. This suggests that HIV integration into intergenic regions or zinc finger genes is associated with a decrease in active HIV expression, which is extremely intriguing (a high priority and clinically relevant area under active investigation in the field). However, there are a few disagreeing observations in the study that seem hard to provide a biological conclusion gained from this study. Overall, this is an elegant study interrogating a mechanistically and clinically important question that should be known to the field.

Major comments:

1. One missing information from Figure 1 is – how long does it take for the timer protein to turn off (to double negative)? This is likely limited by the fact that after 7 days, when the timer protein (R+) has not yet been degraded, the cells may have died of viral cytopathic effect.
2. One interesting finding is that R+ sorted populations have enrichment of HIV integration into ZNF genes, but not double negative (silenced/truly latent) populations. Can the authors elaborate why? For example, can it be that recent HIV activity (R+) affects ZNF gene expression (such as aberrant splicing or transcriptional interference) and provide survival benefit of the cell, while this survival preference is lost if HIV is completely silent (double negative)?
3. Figure 3C and D. Can the authors show the pre- and post-stimulation timer protein expression (FACS, as in 3C) and gene expression (as in 3D) of all 17 clones (if possible), with respective integration site? For example, what's the integration site (gene name, orientation, gene activity, as in 3E) for JO #9, which has no Timer protein expression before and after stimulation, as opposed to JO #10, which has only B+ but not R+ expression (always active)? Clones mentioned in flow cytometry (Figure 3) is not consistent with clones shown in table 1, and cross checking between orientation and integration site makes it hard to interpret the results.
4. JO9 contains HIV integrated into NOSIP (active region), but cannot be induced. Based on previous studies in JLat cell lines, constitutively active HIV-infected cells would likely die of viral cytopathic effects. Those that survived in long term culture, as the authors showed, can either have mutations that reduce cytopathic effect or have additional silencing (such as CpG methylation). Can the authors perform LTR CpG methylation (PMID: 19696893) examination on these cell line clones, instead of correlating with gene expression profiles from other dataset (not the cell lines themselves)?
5. Can the authors elaborate time point chosen for Figure 1I? If the sample matches the data showed in Figure S2, was the R+ population in Figure S2 missing? Is the 'Pos' represents all FP+ population? Can the authors elaborate more why does B+R+ have less Gag DNA?

Minor comments:

1. I'm not sure about 'we were able to concomitantly characterize the whole proviral sequence and proviral expression at single-cell resolution'. Even the HIV-Timer clone is unique in HIV-integration site, the RNA-seq is still a bulk sequencing analysis. I think this statement may be a bit overstating. Please use "for each clone", not "at single-cell resolution".
2. The LPAs treatment promotes latency, and it was proved with the evidence that B+ population underwent decrease while R+ population were increased. Based on the TN population should cover cells that was initially latent, and no or sub-threshold provirus expression and without any detectable Timer

expression. Why do the authors think there was almost no increase in TN population under the treatment of LPAs? Also, based on the genetic and epigenetic similarities between TN and B+ populations, what may result in the differences between these 2 groups under LPA treatments?

3. When the authors mention 'modest increase' 'significant decrease' etc., please make sure all these statements are scientifically rigorous. If an increase is not statistically significant, the authors cannot say "modest increase". There should only be increase/decrease ($p < 0.05$), or no increase/decrease ($p \geq 0.5$). Modest increase has to have $p < 0.05$ but a small effect size (such as 1.1 fold change). Please clarify in each statement.

Reda et al: reply to reviewers' comments:

-We appreciate the reviewers for their precious time in reviewing our manuscript and providing insightful feedback and valuable comments that helped us to improve our work. We have incorporated most of the suggestions made by the reviewers. In the following, we highlight each reviewer's specific comment and our reply to it.

-We would like to ask reviewers to accept one minor change throughout the manuscript: We have removed the hashtag in all JO clone IDs for simplicity.

-All modifications in the manuscript have been highlighted in red.

REVIEWER COMMENTS

Reviewer #1 (Remarks to the Author):

“In this manuscript, Reda et al described the development of a new reporter model to study HIV latency that uses a fluorescent Timer protein that changes its emission spectrum over time. They use this new reporter in two cell lines Jurkat and ThP1 and evaluate expression dynamics, provirus intactness, integration landscape and latency-promoting agents. This new reporter system adds to the tools of other reporters previously used to study latency and add the advantage of the possibility to evaluate latency in cells that revert to a latent state after viral expression, which was impossible till now. In spite of this, there are several concerns with the manuscript as presented that will need to be addressed to have a meaningful impact on the field of HIV.”

Our reply to the comment: We appreciate the reviewer's positive comments. We thank the reviewer for the careful and insightful review of our manuscript. The manuscript has been carefully revised according to the reviewer's comments.

Major concerns:

“1. As stated in the limitations section, the reporter system is only validated in cell lines. It will be important to address whether this reporter system can be use in more relevant primary CD4 T cells and/or macrophages. Otherwise, the title should be change to“HIV-Tocky system to visualize proviral expression dynamics in cell lines”

Our reply to the comment: We thank the reviewer for pointing out this limitation. We added the data supporting the feasibility of infecting primary CD4+ T-cells with HIV_{Timer} construct. This required changes in Fig.1 panel G. We added the description for this result in the text body lines (144-147) under the results section and lines (644-658) under the materials and methods section.

Fig.1G legend:

(G) Representative flow plots from time course infection of Primary CD4+ T cells by HIV_{Timer}. Primary CD4+ T-cells were isolated from PBMCs by negative selection and then activated with Dynabeads Human T-activator CD3/CD28 for 24h. Retronectin-bound virus method was used for infection. Infected cells were followed up for Timer expression until 6 days post-infection. Gating for Timer quadrants in infected cells (lower panel) was performed according to un-infected controls (upper panel).

“2A. “Figure 1. The MOI for the NGFR virus is above 1 and that can lead to multiple infections per cell as shown in sup Fig 2C. This could lead to misinterpretation of some of the results, in particular with bulk analysis”. **2B.** It is also not clear whether all Blue/Red cells are NGFR positive. **2C.** It will be important to demonstrate that in each population generated (TN, B, R, BR) contains intact proviruses as it is well known that even with replication-deficient HIV constructs, deletions can occur. In particular when other promoters and fluorescent proteins are introduced. **2D.** A transcript control (like NGFR) should be included in parallel to Fig 1I to demonstrate that only the HIV promoter is silenced but not the NFGR. **2.F** Also, it is not clear whether the analysis was done in Jurkat or THP1.”

Our reply to the comment: We thank the reviewer for raising such important points and concerns.

2.A. The data shown in Fig.S2C (updated to Fig.S3C) included proviral load calculation from sorted Timer fractions of Jurkat cells infected with HIV_{Timer} construct. Cells infected with this construct lack NGFR expression cassette. We agree with the reviewer's opinion that some infected cells can carry more than one copy of the virus, but in all three fractions, PVL didn't reach 200% meaning that less than 2 copies per cell is mostly achieved in our infection system. In the same context, we would like to show our cell-associated viral DNA calculation from HIV_{TNGFR} infection into Jurkat cells for the reviewer using Jurkat cells infected with HIV_{TNGFR} shown in Fig.1D in addition to Fig.S3C. We extracted DNA from bulk infected cells (Fig.1D) at 3 dpi and

quantified cell-associated viral DNA as described under the Digital droplet PCR section of the materials and methods.

Shown in a 1D dot plot, are droplet results for *Gag* and *Albumin*.

Sample	Alb (copies/uL)-FAM	Gag (copies/uL)-HEX	Cell-associated viral DNA (%)
Jurkat-HIV _{TNGFR} -3dpi	56.9	25.3	89

Cell-associated viral DNA% was calculated at 89%

2.B. We, here, would like to show the reviewer the results of the blue and red populations' NGFR expression (Example: 3dpi time point)

2. C. We added the data for the amplification of near-full-length single HIV-1 genome from the same samples shown in Fig. S2 (now updated to Fig. S3). We described that in the results section (lines 171-174) and the methods section (lines 717-731).

G

Fig.S3G legend:

(G) Pie charts reflecting the proportion of defective and intact proviruses presented in (C); n= total number of provirus sequences analyzed.

2. D. As advised, we have checked NGFR mRNA expression levels across different Timer populations. We have added the results to Fig. S2B. We modified the text description for this result in lines (159-163) under the results section, and lines (685-702) under the methods section.

Fig.S2B legend:

(B) Total RNA isolated from each Timer population of Jurkat-infected cells with HIV_{TNGFR} used in Figure 1I was subjected to SYBR green RT-qPCR analysis. NGFR mRNAs were quantified relative to cellular GAPDH and fold change in NGFR expression was calculated relative to its level in TN population. (n = 2 biologically independent experiments, mean ± SD).

2. F. We have added a label on the validation graph of Fig.1I and its legend denoting that the experiment was performed in Jurkat T-cells. A description of the experiment is included in the text (Lines 152-159).

Fig. 1I legend:

(I) Total RNA isolated from each T-cell population of Jurkat-infected cells with HIV_{TNGFR} was subjected to SYBR green RT-qPCR analysis. Unspliced *Gag* (US) in teal and multiply-spliced *tat/Rev* (MS) in lime green. HIV-1 mRNAs were quantified relative to cellular 18s rRNA. (n = 2 biologically independent experiments, mean ± SD).

“3. In figure 2S, the 2-LTR data should also be shown and not as data not shown if it has been done. Even it is a replication-deficient provirus, 2LTR and 1LTR can still form. Furthermore, the HIV DNA analysis performed does not distinguish between integrated and non-integrated HIV DNA.”

Our reply to the comment:

(i) We understand the reviewer's concerns. We have improved our detection system for 2-LTR DNA and included the updated findings in Fig. S3D, E, and F. Overall the time points we checked, 2-LTRc contributed to less than 1% of all detected *Gag* copies. We have updated the manuscript describing this result in lines (167-170).

Fig. S3D, E, and F legend:

(D) 1D dot plot showing droplet fluorescence intensity on the y-axis and each sample tested on the x-axis for 2-LTRc (upper panel) and *Gag* gene (lower panel). (E) 2-LTRc dynamics by quantifying copies/uL from ddPCR data shown in (D) starting at 12 h post-infection until 3 dpi. Copies/uL per time point were divided by copies retrieved on 1 dpi and plotted as a percentage. (F) Pie charts showing the percentage of 2-LTRc to total *Gag* per time point for data shown in (E).

(ii) We understand the point of not distinguishing integrated and non-integrated HIV DNA shown in Fig. S3C. We improved the clarity of the figure by replacing the term proviral load percentage with Cell-associated viral DNA% and updated that in the main text in lines (165-167).

C

Fig. S3C legend:

(C) Cell-associated DNA for each Timer population was calculated by the following formula (copy number of HIV-1 gag DNA)/[(copy number of Albumin)/2] x100; pos is assay positive control.

“4. Figure 2. Because of the high multiplicity of infection, the integrating analysis in the 4 populations may be misleading as multiple integrations may be in an individual cell. This should be either address experimentally or mention as another caveat of the study.”

Our reply to the comment:

We thank the reviewer for this comment. We have discussed our findings in our reply to the previous comment 2A. Additionally, we added this point to the limitations of our study in lines 605-607 of the manuscript text.

“5. Figure 3. It is unclear which clones are from HIVTimer or HIVTNGFR. It should be demonstrated that each one has 1 copy per cell and that the provirus is intact. It will be beneficial to include the integration site for each clone in the figure C for clearness. Changes in NGFR expression concomitant with activation should be included to evaluate that the increased in blue/red is associated with changes in HIV transcription and not general transcription.”

Our reply to the comment:

-We appreciate the reviewer’s detailed comment on these important points.

(i) We have added one column in Table 1 showing the virus construct used to generate each clone. Also, this has been described in the table legend.

(ii) We have added one column demonstrating the percentage of cell-associated viral DNA in each clone. Using our DNA-capture-seq pipeline, we screened all the clones for intactness and showed that in Table 1 as well.

(iii) We modified Fig. 3C for better clarity by including the host gene name of integration for each clone.

Fig.3C legend:

(C) Flow cytometry plots showing Timer FP expression denoting provirus expression in selected single integration Timer clones. Jurkat Timer clones' basal expression (upper left panel); provirus expression after T-cell stimulation for 24 h (lower left panel); THP-1 clones' basal expression (upper right panel) and THP-1 provirus expression after TNF- α stimulation for 24 h (lower right panel). The host gene of HIV-1 provirus integration in each clone is shown under each clone ID. Plus or minus denote the same or convergent orientation of provirus integration relative to the host gene, respectively.

(iv) We added the flow-cytometry plots showing changes in NGFR expression before and after Timer clones' stimulation in Fig.S7. We used the same THP-1 clones shown in the original figure (Fig. 3C) for that purpose as an example. We modified the text description of that part in lines (322-326).

Fig. S7. Changes in NGFR expression with Timer clones' activation. Examples from THP-1 Timer clones (shown in Figure 3C) generated with HIV_{TNGFR} construct infection are shown for basal expression and after stimulation with TNF- α for 24 h.

“6. Figure 4. The use of LPAs is interesting, but it will be important to address if the reduction on transcription is just associated with HIV transcription or a shutdown of all transcription. For example, what are the levels of NGFR or other cellular protein? Also, Figure E is missing any statistical analysis.”

Our reply to the comment:

-We thank the reviewer for this suggestion.

(i) Clone JO19 which we used for demonstrating the LPA effect is generated by HIV_{Timer} construct and lacks NGFR expression. To examine the status of cellular gene expression, we used GAPDH as a cellular control to distinguish the shutdown of HIV transcription from general cellular transcription. We have added results from this experiment in Fig. S14 and described it in the text body in lines (419-424).

Fig. S14 Legend:

Changes in cellular transcription in comparison to HIV transcription by LPA treatment

Total RNA isolated from Jurkat Clone JO19 cells treated with DMSO or LPA (Levosimendan at 10 μ M, and Triptolide at 10 nM) at 12 h post-treatment (shown in Figure 4 D) was subjected to SYBER green RT-qPCR analysis. GAPDH mRNAs were quantified relative to cellular 18S rRNA

and the fold change in GAPDH expression was calculated relative to its level in Clone JO19 treated with DMSO. Value from JO19 stimulated with TNF- α for 48h (before LPA treatment initiation) is also shown. (n = 2 biologically independent experiments, mean \pm SD).

(ii) We apologize for missing the statistical analysis of Fig. 4E. We have modified the figure to include the statistical significance data.

Fig. 4.E. legend:

(E) Transitions in each Timer population from zero point until 12 h post-treatment are graphed by expression percentage obtained from flow cytometry analysis of each Timer population against time under Levosimendan treatment (left panel) or Triptolide (right panel) with 3 stated different concentrations in comparison to DMSO treatment (n = 3 biologically independent experiments, mean \pm SD). Statistical significance in (B; right panel) and E was assessed by Paired t-test; ns= not significant, * $<$.05, ** $<$.01, *** $<$.001.

Minor concerns:

1. Line 381 is missing a part of the sentence.

Our reply to the comment: We have edited the text (line 538) to complete the sentence.

Reviewer #2 (Remarks to the Author):

“In this manuscript, Reda et al. describe a new system to track HIV proviral integration and subsequent establishment of latency. This ingenious system uses a timer protein that switches from "blue" to "red" fluorescence over time. The manuscript is well detailed, relevant for the HIV latency/cure field, and well articulated. The authors describe openly known limitations which is great for scientific rigour.”

Our reply to the comment: We appreciate the reviewer’s positive comments. We thank the reviewer for the careful and insightful review of our manuscript. The manuscript has been carefully revised according to the reviewer’s comments.

Major Comments:

“1-(Lines 198-199) It is not clear in Fig S4B that there are differences in distances from the TSS. The distances seem to distribute evenly, with different count number.”

Our reply to the comment:

We thank the reviewer for raising this point. We have modified the figure (updated to Fig. S5B) for better clarity to show the median distance of all integration events from TSS in each Timer population separately as showing overlapped values in the previous figure masked the difference. Median values for Timer Red integrations are more distant from TSS in comparison to other Timer populations’ median distance but the difference was not statistically significant.

Fig. S5B legend:

(B) Histograms showing the distribution of ISs within each Timer population in relation to the chromosomal distance to the most proximal TSS; the median value is shown for each population. Statistical significance was calculated by Steel-Dwass’s multiple comparison test.

“2- (Lines 224-225) Although statistically significant, which is easy to achieve with the high number of replicates, it is not clear that the R+ integration in ZNF genes is biologically relevant, the authors could perhaps use other controls in the figure (one showing little integration, and another more abundant. It is hard to estimate and normalize "ZNF genes" as they are clustered as a family vs others, as their abundance is not stated.”

Our reply to the comment:

(i) We appreciate raising this important point. We have added the total number of integrations in genes for each Timer population, from which we calculated the percentage of ZNF abundance in Fig. 2F and its related legend.

Fig. 2.F legend:

(F) The frequency of unique ISs within ZNF genes was calculated from total genic integrations in each Timer population, numbers in white on each bar denote the percentage of ZNF genes calculated from the total number of gene integrations shown as (n=). For figures (B, E, and F); Statistical significance was assessed by Fischer's exact test. *<.05, **<.01, ***<.001.

(ii) Next, we would like to discuss on the biological relevance of this finding:

-ZNF genes are reported to be associated with repressive chromatin marks and to support long-term persistence for HIV-1 integrated proviruses. Owing to that, we looked into the R+ proviruses distribution in these genes to verify the observation we got from ChIP-seq data showing that the R+ proviruses tend to integrate in close proximity to repressive histone marks.

-We improved our description for that part in the manuscript for better clarity (lines 261-265).

-Additionally, we would like to show the reviewer some preliminary data from extended analysis from ZNF integrations in R+ population:

-----IGV profile for integrations in ZNF clusters in chromosome 19 previously described in (PMID: 37698927). In this mentioned study, authors traced the clonal dynamics of two intact proviruses integrated in ZNF genes before and after chemotherapy in an elite controller. Of these two clones, one has persisted, and the other has expanded under chemotherapy.

ZNF Integrations in R+ population

IGV profile of ZNF integrations in chromosome 19 from sorted Timer R+ population. Green square denotes the specific gene of integration in our data set.

----We additionally sorted out TN and R+ populations and followed the dynamic changes in provirus expression by Timer-FP until 6 days post-sorting without any treatment or induction R+ proviruses tend to keep their silenced behavior (11% remained R+ and 85% went into latency). On the other hand, TN proviruses tended to reactivate. Here, we show the preliminary result we got from this experiment which requires further validation with sorting higher cell numbers and following for longer time points.

Provirus expression dynamics in sorted TN and R+

Sorted Timer populations (TN and R)HIV_{TNGFR} infected Jurkat T cells shown in S2A were followed up for changes in provirus dynamics until 6 days post sorting. FACs plots show changes in TN population (1-6 days) post sorting, while R+ were checked once at 6 days post sorting.

(iii) Combining all these results, we hypothesize that R+ proviruses harbor distinct features of proviral latency (including ZNF integrations), while TN is a heterogeneous group of proviruses that can readily express stochastically because their integration environment doesn't hinder that, unlike R+ proviruses.

“3- (Lines 270-271). The authors state that their clones JO#10 and JO#19 results contradict the interference phenomenon. (i) authors should further introduce the phenomenon for a wider audience (ii) it is perhaps strong to arrive such conclusions with only two clones. It could perhaps be achieved with 10+ clones.”

Our reply to the comment:

-We appreciate the reviewer's comment and concern about this point.

-There was a misunderstanding about what we mean in these 2 lines. In our studied clones, we experienced a low basal proviral expression in clone JO10, while the provirus is integrated in an active transcribing gene (*HNRNPM*). On the contrary, we observed a high basal proviral expression in clone JO19 while its provirus is integrated into a silenced gene (*TSBP-1*). Owing to the opposite integration of the provirus in clone JO10, we attributed this phenomenon partially to transcriptional interference. However, we agree that further experiments are required to prove or deny that completely.

-We have modified the text to include more explanation for TI for a wider audience and also, we improved the clarity of our explanation about the provirus expression we monitored in each clone. Text modification in lines (342-352).

“4-In Fig 3C, The JO#19 clone shows an increase of R+B+ cells vs JO#10, wouldn't that mean that it is somewhat independent of the TNF α stimulation? Shouldn't the timing of TIMER R+ be the same for both clones (with normalization to the 4% positive before stimulation?)”

Our reply to the comment:

-We thank the reviewer for this comment. We think that the differences in provirus expression patterns between clones are related to the provirus integration landscape and factors affecting it, which would explain the high basal proviral expression in one clone (JO19) but not the other (JO10) and are independent of TNF- α stimulation. It reflects how the integration environment is permitting the provirus expression.

-This also controls the timing of R+ appearance and causes the difference observed. Inter- and intra-clonal variability in basal proviral expression has also been reported before by Jordan et al., 2001. (Ref 40 in the manuscript).

“5- The formation of defective proviruses should not be dependent on Env as the authors state. Although not clear what drives the high number of defective proviruses in vivo, it is likely a selection via CD8/NK cells that kills and cleans any cells expressing HIV (including full proviruses). The sentence should be revised.”

Our reply to the comment:

-We appreciate pointing out this important point by the reviewer. We revised the sentence and updated the text to “We used single-infection-cycle HIV molecular constructs to limit the appearance of defective proviruses.” (lines 488).

Minor Comments:

1. Line 85 the word spontaneously should be more characterized, later the authors describe the time it takes for Timer protein to change its spectrum. The text would benefit from a better description in this section.

Our reply to the comment:

-We have modified the text to include more information on the spontaneous change of the Timer-FP fluorescent spectrum at line 85.

-Updated text “This Tocky model uses a fluorescent Timer protein that spontaneously changes its emission spectrum from blue to red upon maturation of blue chromophore by oxidation

30,31.”

2. It is not clear to me what are the differences between the data of Figure 1C/D and Figure S1, please clarify in the text.

Our reply to the comment:

-We understand the reviewer's query. Fig.1C represents the time course of infecting Jurkat T-cells with HIV_{Timer} construct. Fig.1D represents the time course of infecting Jurkat T-cells with HIV_{TNGFR} construct. Due to the busy figure, we only explained that in the figure legend.

-Fig. S1(now updated to Fig.S2A) represents infection of Jurkat T-cells with HIV_{TNGFR} construct with subsequent Timer fractions sorting and purity check for each population.

3-The authors describe that they sorted different populations for B+, B+R+ and R+ in lines 150 to 154. It should be clear in the text the time post-infection where that was performed.

Our reply to the comment:

We appreciate the reviewer's comment. We have moved that explanation into main text lines (155-157)

4-Authors state that no 2-LTR circles were detected with data not shown. Such data (and proper controls, should be provided). Although at later time points 2LTR circles tend to disappear, explaining such observation it is important for the scientific rigor to present such data.

Our reply to the comment:

We appreciate the reviewer's raising of this important point. We kindly ask the reviewer to check our reply concerning this point to reviewer #1 (Major comment 3).

5-(Line 370) Please clarify the usage of the word asymmetric, do the authors mean distinct/different/independent?

Our reply to the comment:

We have updated the text for better clarity. Now updated to line 527: "Our data describe two different layers of latency with differential regulatory determinants at which HIV-1 provirus reactivation and silencing appear to be different phenomena, with the former being more stochastic and the latter being more controlled."

6- The authors compare the "shock and kill" and "lock and block" strategies, describing a higher viability for the "lock and block" strategy. Wouldn't the transcription stochasticity (very well described in the manuscript), apply in a "lock and block" strategy as well?

Our reply to the comment:

-We appreciate the reviewer's comment and concern. The purpose of our manuscript is to propose and validate the HIV-Tocky system to capture provirus dynamics. Because of its uniqueness in

capturing the silencing dynamics, we investigated its potential to capture the provirus silencing dynamics more sensitively in comparison to previous fluorescent models.

-We again understand the reviewer's point of view that comparing the efficiency of LRAs or LPAs is out of the scope of this study, and thus we modified the text to be more focused on our study aim. Modified text (lines 560-574)

7-The THP-1 model should be described in the limitations of the study, especially because the authors did not differentiate the cells into "macrophage like" cells with PMA or other methods. Monocytes are not a "canonical" target of HIV-1, as they do not have the CD4 receptor.

Our reply to the comment:

We have modified the text describing our study limitations to include that. Now updated to lines (602-604).

8- Please provide amounts of DNA, cell number, and plate type/area for virus production and infection.

Our reply to the comment:

We have modified the text providing this information. Now updated to lines (630-631 and 639-640).

9-Please show Data referred to in line 529.

Our reply to the comment:

We modified the figures to include this data, now updated to Figure S15.

Reviewer #3 (Remarks to the Author):

“Reda et al. developed a novel HIV-Tocky system which can identify active versus inactive HIV gene expression dynamics by measuring dual fluorescence emission spectrum shifting (Timer protein). By using this HIV-Tocky system, the authors applied interrogated HIV integration heterogeneity, clonal expressions, and tested latency stability under latency promoting agent (LTA) treatments. The HIV-Tocky system is elegantly designed to capture temporal dynamic changes after HIV reporter virus infection and the fan-shape changes of the Timer fluorescence proteins reflected the latent proviruses. The authors performed HIV integration sites analysis of active (B+) versus inactive (R+). In Figure 1, the authors showed the dynamic changes of Timer protein fluorophore over time in two cell lines, Jurkat and THP-1. In Figure 2, the authors examined the HIV integration site landscape of active (B+) versus inactive (R+) HIV infection. In Figure 3, the authors established long term culture of these HIV-timer clones. Among these 17 clones, 2 (11%) have HIV integrated into

silenced genes while the remaining 15 have HIV integrated into actively transcribed genes. In Figure 4, The authors used this HIV-timer system to evaluate drug effect as a tool for potential clinical implications for drug testing. The major strength of this manuscript is the elegant use of timer protein and the rigorous analysis of integration site and near fulllength proviral genome sequencing. The striking finding is that inactive HIV infection (R+) seems to have more integration into intergenic regions associated with repressive chromatin marks and integration into zinc finger genes. This suggests that HIV integration into intergenic regions or zinc finger genes is associated with a decrease in active HIV expression, which is extremely intriguing (a high priority and clinically relevant area under active investigation in the field). However, there are a few disagreeing observations in the study that seem hard to provide a biological conclusion gained from this study. Overall, this is an elegant study interrogating a mechanistically and clinically important question that should be known to the field.”

Our reply to the comment: We appreciate the reviewer’s positive comments. We thank the reviewer for the careful and insightful review of our manuscript. The manuscript has been carefully revised according to the reviewer’s comments.

Major comments:

“1. One missing information from Figure 1 is – how long does it take for the timer protein to turn off (to double negative)? This is likely limited by the fact that after 7 days, when the timer protein (R+) has not yet been degraded, the cells may have died of viral cytopathic effect.”

Our reply to the comment:

-We appreciate the reviewer's comment.

-In this study, we haven’t measured the half-life of Timer Blue or Red fluorescence. The half-lives stated in our manuscript are as per identified by Bending et al., 2019 (Ref 30) in murine CD4+T-cells. We agree with the reviewer’s comment that owing to the long half-life of Timer red protein, 7 days is not long enough to monitor the shutdown of transcription in all proviruses. Most expressing cells will decrease over this period not by the decay of Timer Red protein but by CPE.

“2. One interesting finding is that R+ sorted populations have enrichment of HIV integration into ZNF genes, but not double negative (silenced/truly latent) populations. Can the authors elaborate why? For example, can it be that recent HIV activity (R+) affects ZNF gene expression (such as aberrant splicing or transcriptional interference) and provide

survival benefit of the cell, while this survival preference is lost if HIV is completely silent (double negative)?”

Our reply to the comment:

-We thank the reviewer for discussing this important point with us.

-As we have shown in our ChIP-seq results; R+ proviruses had a preference to integrate in close proximity to repressive histone marks when compared to other Timer populations including TN (double negative). In that sense, we queried the distribution of R+ proviruses integrations in ZNF genes known to be associated with repressive histone marks, in comparison to other Timer populations as additional proof of the initial finding we observed from ChIP-seq results.

-We improved our description for that part in the manuscript for better clarity (lines 261-268).

“3. Figure 3C and D. Can the authors show the pre- and post-stimulation timer protein expression (FACS, as in 3C) and gene expression (as in 3D) of all 17 clones (if possible), with respective integration site? For example, what’s the integration site (gene name, orientation, gene activity, as in 3E) for JO #9, which has no Timer protein expression before and after stimulation, as opposed to JO #10, which has only B+ but not R+ expression (always active)? Clones mentioned in flow cytometry (Figure 3) is not consistent with clones shown in table 1, and cross checking between orientation and integration site makes it hard to interpret the results.”

Our reply to the comment:

We understand the reviewer's point and apologize for the confusion.

We revised to figure to improve clarity by Changing the following points:

- (i) Fig. 3C: we included information on the gene of integration, orientation, and gene activity.

Fig.3C legend:

(C) Flow cytometry plots showing Timer FP expression denoting provirus expression in selected single integration Timer clones. Jurkat Timer clones' basal expression (upper left panel); provirus expression after T-cell stimulation for 24 h (lower left panel); THP-1 clones' basal expression (upper right panel) and THP-1 provirus expression after TNF- α stimulation for 24 h (lower right panel). The host gene of HIV-1 provirus integration in each clone is shown under each clone ID. Plus or minus denote the same or convergent orientation of provirus integration relative to the host gene, respectively.

- (ii) Starting from Fig. 3C onwards, our study focused on generated clones with a single integration event. Thus, we performed further analysis on 10 Jurkat clones (stated in Table 1) each of which harbors only one provirus integrated.
- (iii) We included pre- and post-stimulation flow cytometry plots for the rest of the single integration Jurkat clones in Figure S8.

Fig. S8. Flow cytometry plots showing Timer FP expression denoting provirus expression in selected single integration Timer clones. Jurkat Timer clones' basal expression (upper left panel); provirus expression after T-cell stimulation for 24 h (lower left panel); THP-1 clones' basal expression (upper right panel) and THP-1 provirus expression after TNF- α stimulation for 24 h (lower right panel). The integration site of HIV-1 provirus in each clone is shown under each clone ID. Plus or minus denote the same or convergent orientation of provirus integration, respectively.

- (iv) We apologize for the incorrect information about JA3 clone inducibility in the initially submitted manuscript. We have updated the information in Table One according to the current clone status.
- (v) For JO9, we have updated the figure including (gene name, orientation, and gene activity), but we have not performed mRNA-seq for this clone as we have not monitored any expression from this clone and thus we didn't expect an effect on the host gene expression. We additionally provided information on the expression of the

gene of integration in Clone 9 from the public data set as in 3E in Fig.S10.

Fig. S10. Integration environment of Timer Clone JO9

(A) RNA-seq and ChIP-seq data sets of Jurkat T-cell lines were blotted by UCSC Genome Browser (<https://genome-asia.ucsc.edu>) and shown in respect of integration gene of Timer clones JO9. Schematic for the position and directionality of provirus integration in this clone is demonstrated at the bottom.

“4. JO9 contains HIV integrated into NOSIP (active region), but cannot be induced. Based on previous studies in JLat cell lines, constitutively active HIV-infected cells would likely die of viral cytopathic effects. Those that survived in long term culture, as the authors showed, can either have mutations that reduce cytopathic effect or have additional silencing (such as CpG methylation). Can the authors perform LTR CpG methylation (PMID: 19696893) examination on these cell line clones, instead of correlating with gene expression profiles from other dataset (not the cell lines themselves)?”

Our reply to the comment:

We thank the reviewer for this fruitful suggestion and for bringing up this important reference. We have performed Bisulfite sequencing to detect LTR CpG methylation. We have included the results in Fig. S11 and described them in the main text (lines 353-366 and 747-763).

Overall, we have analyzed the 5' LTR CpG methylation in 3 of Jurkat clones, two of them were non-inducible (JO9 and JA8), and one was inducible as a control (JA3). JO9 showed 1.7-fold higher CpG methylation than the inducible clone A3. However, Clone JA8 showed the lowest methylation pattern. Our findings suggest that 5'LTR CPG methylation may play some role in promotor silencing but is not a major determining factor.

Fig. S11. CpG methylation profile of the 5' LTR in Jurkat Timer clones JO9, A3 and A8. Every row is a single genome sequence. Black circles denote methylated CpG residue, open circles denote un-methylated CpG residues (mCPGs). Numbers above each graph denote CpG islands' location within the 5' LTR region. Position 359 is NFAT/NFkB- binding site and positions 382,393, and 408 are Sp1-binding sites.

“5. Can the authors elaborate time point chosen for Figure 1I? If the sample matches the data showed in Figure S2, was the R+ population in Figure S2 missing? Is the ‘Pos’ represents all FP+ population? Can the authors elaborate more why does B+R+ have less Gag DNA?”

Our reply to the comment:

-We thank the reviewer for these questions.

-First, the samples in Fig.1I match the samples shown in Fig. S2. The Sorting time for each population is mentioned in the text lines (155-157).

-Second, in Fig. S2C (now updated to S3C); pos means positive control. We have updated the figure legend to clarify that.

-Third, we updated the figure to include the values (cell-associated viral DNA) for each population. These results are obtained from Jurkat-infected cells with HIV_{Timer} construct which lacks a marker of infection (NGFR), in this case, we expect cell-associated viral DNA to be detected at similar levels in all sorted populations except for TN which is a mixture of infected and un-infected cells. The difference in values between the B+ and B+R+ population is very narrow; however, we attribute that to the higher provirus copies in B+ (expressing) population which is then more prone to be cleared by CPE before reaching B+R+ phase.

C

Fig. S3C legend:

(C) Cell-associated DNA for each Timer population was calculated by the following formula (copy number of HIV-1 gag DNA)/[(copy number of Albumin)/2] x100; pos is assay positive control.

Minor comments:

1. I'm not sure about 'we were able to concomitantly characterize the whole proviral sequence and proviral expression at single-cell resolution'. Even the HIV-Timer clone is unique in HIV-integration site, the RNA-seq is still a bulk sequencing analysis. I think this statement may be a bit overstating. Please use "for each clone", not "at single-cell resolution".

Our reply to the comment:

We thank the reviewer for this correction. We have modified the text accordingly (line 369)

2. The LPAs treatment promotes latency, and it was proved with the evidence that B+ population underwent decrease while R+ population were increased. Based on the TN population should cover cells that was initially latent, and no or sub-threshold provirus expression and without any detectable Timer expression. Why do the authors think there was almost no increase in TN population under the treatment of LPAs? Also, based on the genetic and epigenetic similarities between TN and B+ populations, what may result in the differences between these 2 groups under LPA treatments?

Our reply to the comment:

We thank the reviewers for discussing these important points.

- (i) We have been able to monitor an increase in TN population in comparison with Levosimendan treatment in comparison to DMSO but not in the case of Triptolide treatment. We attribute that to the difference in the mechanisms of actions of both drugs; where Levosimendan acts by Tat degradation and thus acts more specifically on the provirus dynamics in comparison to Triptolide which is an inhibitor for NF- κ B activity.
- (ii) In answer to the second question, we expected the halting of expression of proviruses in the B+ quadrant (reflected by their decrease) and of TN proviruses (reflected by their increase). We have been able to see the first effect but not the second. We attribute that to the limitation of flow cytometry analysis gating owing to the adjacent TN and R+ populations, so we may have been a bit underestimating the TN population.

3. When the authors mention 'modest increase' 'significant decrease' etc., please make sure all these statements are scientifically rigorous. If an increase is not statistically significant, the authors cannot say "modest increase". There should only be increase/decrease (p <0.05), or no

increase/decrease ($p \geq 0.5$). Modest increase has to have $p < 0.05$ but a small effect size (such as 1.1-fold change). Please clarify in each statement.

Our reply to the comment:

We thank the reviewer for pointing this out. We apologize for any inaccuracy. We have revised the manuscript to modify these descriptions. (Lines 241, 245, and 339).

REVIEWERS' COMMENTS:

Reviewer #1 (Remarks to the Author):

The authors have addressed my previous comments.

Reviewer #2 (Remarks to the Author):

In this manuscript, Reda et al. describe a new system to track HIV proviral integration and subsequent establishment of latency. This ingenious system uses a timer protein that switches from "blue" to "red" fluorescence over time. The manuscript is well detailed, relevant for the HIV latency/cure field, and well articulated.

New comments:

The Article revision did a great job replying to all of my concerns and the new data and improvements fully align the data, interpretation to the discussion and conclusions.

Reviewer #3 (Remarks to the Author):

The authors have sufficiently addressed my previous questions.